# The Role of the Interface and Interface Management in the Optimization of BIM Multi-Model Applications: A Review

**Nawal Abdunasseer Hmidah [1,*], Nuzul Azam Haron [1], Aidi Hizami Alias [1], Teik Hua Law [1], Abubaker Basheer Abdalwhab Altohami [1] and Raja Ahmad Azmeer Raja Ahmad Effendi [2]**

1   Department of Civil Engineering, Faculty of Engineering, University Putra Malaysia (UPM), Serdang 43400, Malaysia; nuzul@upm.edu.my (N.A.H.); aidihizami@upm.edu.my (A.H.A.); lawteik@upm.edu.my (T.H.L.); gs50517@student.upm.edu.my (A.B.A.A.)
2   Department of Industrial Design, Faculty of Design & Architecture, University Putra Malaysia (UPM), Serdang 43400, Malaysia; azmeer@upm.edu.my
*   Correspondence: gs57074@student.upm.edu.my or nawal.aljahmi@yahoo.com

**Abstract:** This review targets the BIM interface, the BIM multi-model approach, and the role of employing algorithms in BIM optimization to introduce the need for automation in the BIM technique, instead of complicating manual procedures in order to reduce possible errors. The challenge with adopting BIM lies in the limiting ability of computer-aided design (CAD) to generate a read-able and straightforward Revit by BIM, requiring the homogeneous data format to be generalized better and maintain a super data mod. Furthermore, the communication and management inter-face (CMI) faces some shortcomings due to limitations in its ability to recognize the role of the interface during the project construction phase. This review demonstrates several proposals to simplify the interface, in order to facilitate better communication amongst participants. The industry foundation class (IFC) model requires a new technique to unlock the potential future of intelligent buildings using the BIM multi-model approach integrated with the Internet of Things (IoT). Trials conducted to enhance the BIM model lack advanced methods for optimizing cost, energy consumption, labor, material movement, and the size of layout of the project, by utilizing heuristic, metaheuristic, and k-mean algorithms. The enhancement of BIM could involve algorithms to achieve better productivity, safety, cost, time, and construction frameworks. The review shows that some gaps and limitations still exist, especially considering the potential link between BIM and building management system (BMS) and the level of influence of the BIM-IoT prototype. Future work should find the best approach to solve facility management within the dynamic model, which is still under investigation.

**Keywords:** BIM; management interface; BIM multi-model; BIM-BMS system; optimization

## 1. Introduction

BIM is a project-improving tool that globally provides a revolutionary platform for design, construction, maintenance, operation, and improvement in various fields for the rehabilitation, retrofit, and redevelopment of existing assets in the built environment. Another helpful definition considers BIM as a methodology that combines several processes and tools to improve projects and overall construction outcomes [1].

BIM is a paradigm that shifts the inefficient 2D drawing and processes and practices of documentation towards much more precise model-centric processes and practices. Re-searchers consider BIM as an integrated information system that effectively assimilates the organizational functions and processes of project delivery. BIM is an inclusive term that can be defined in diverse ways; however, the most typical definition states that BIM is software used to create value and promote collaboration in the entire lifecycle of an asset, using underpinning theories by collating and exchanging 3D models [2].

BIM has reached an exciting stage, as many built environment stakeholders are cur-rently using or considering using it. As reported in 2017, 86% of UK respondents expect to

adopt BIM for their projects [3]. BIM adoption level varies from one country to another, depending on the size and complexity of the projects.

At its inception, building information modelling (BIM) was associated with using 3D modeling with the availability of various software tools and techniques. Traditionally, although 3D construction models had been integrated for additional measurements of time and cost, they were found inadequate in terms of including all the project-specific details necessary for a building project. On the other hand, BIM is equipped with enough technicalities to create virtual 3D models by integrating relevant information, and simultaneously granting project participants a better understanding of the project phases [4]. Concerning expanding models, Ivson et al. [5] developed several models that were utilized to represent models and several corresponding sub-models to serve different operations simultaneously.

Accordingly, the philosophy of the multi-model approach has emerged to collate data from various sources with different formats into a single exchangeable resource [6], which can be characterized as object-oriented [7].

The objective of BIM is to create accurate, reliable, complementary, and replaceable information for the construction of buildings [8]. These objectives can only be attained by implementing interoperability and parametric (adjusting variables) behavior. Eastman et al. [9] defined BIM as a technology equipped with a set of processes that aim at producing, communicating, and analyzing building models. BIM is widely considered a significant factor in the construction industry. BIM describes an integrated model-based view of a facility's lifecycle, including design, planning, and construction, as well as operation and maintenance (O&M) [10].

Recently, BIM has been adopted in various types of projects, and in projects that require dynamic data exchange amongst multiple actors with information aggregation, such as designing a project, running software, handling data, revising all or parts of the project [11], and improving the efficiency of construction [12].

Briefly, the BIM model is described as a mixture of graphical and non-graphic data that can communicate throughout specific data-exchange formats.

Recently, it has been observed that BIM applications are expanding to many fields, owing to the introduction of 3D geometric models and 3D coordination [13]. These applications go beyond architecture and engineering, to cover and initiate a strong motive for homeowners, facility managers, contractors, and fabricators [14]. The project focuses on BIM adoption, provided by utilizing automation in the modeling process. This modeling improves communication and accuracy among various parties throughout, exchanging views and reducing the errors in the coordination of building activities [15]. BIM applications plan, design, build, construct, operate, and reduce energy consumption [16]. These developments were not applicable to certain countries; they are, instead, applied to all countries, sharing the same principles of integration of BIM and building energy management (BEM) in a single tool [17].

The fundamental contributions of BIM are in energy-related matters, simulations, and information, which can be described as involving the automation of energy to better present output in order to enhance storage and organizational capabilities concerning new-building data. The other contribution of BIM concerns the facilitation of output presentations in energy management systems [18]. Similarly, [19] studied a conceptual framework for a BIM-based energy management support system (BIM-EMSS) by developing a real-time energy simulation using eQuest. Based on the determination of [18], visualizing the geometric data in BIM could allow the user to monitor the real-time energy performance of different zones in a building.

Other benefits of BIM applications include storing, monitoring, and organizing energy-related information in real-time energy systems. The system can generate information related to home energy consumption and how to relate activities to environmental temperature and the degree of occupancy. The adoption of BIM models in real-time energy monitoring systems was explained by Alahmad et al. [20], who proposed a combined

system that uses a hardware component system and a software system. Woo et al. [21] reported other BIM applications in a building equipped with sensors that provide real-time data to BIM models using a standard schema to facilitate processing the data related to sensors and actuators. The other important application of BIM is performed by linking existing libraries, where a great amount of information about the thermal conductivity properties is available. The life cycle assessment of a building can be estimated better by integrating CAD and BIM. This link provides information about optimizing the building envelope or sizing the HVAC system [22].

## 2. BIM and Interface Management Flowchart

The flow of topics in this review is outlined in Figure 1. The purpose of this flowchart is to guide readers for easy access to the topics that are included in this review, and to present the contents in a structured fashion.

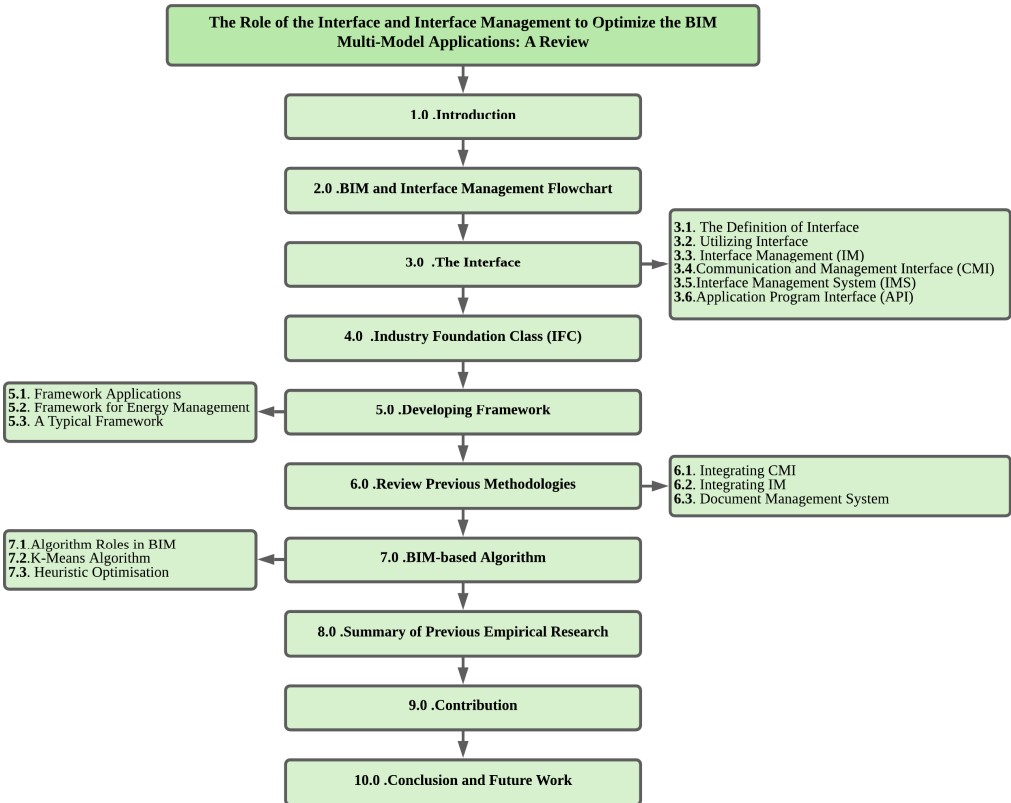

**Figure 1.** Contents flowchart.

## 3. The Interface

### 3.1. The Definition of Interface

The definition of the interface has been developing since 1967, when Wren [23] denoted that the interface is the contact point or set of points (surface) between two independent systems to achieve a better, more extensive and unified system. However, as time progresses, other researchers have been proposing numerous definitions of the interface.

Lin [24] expanded the role of the interface to include cases related to different opinions, such as schedule, cost, technical areas, and the space between systems. Interface management (IM) is another element of interface to address the challenges of managing complex capital projects to face the rising complexity due to globalization and the geographical distribution of various cultures [25]. Shen et al. [26] have provided another depth of the interface by correlating the interrelation and interaction among different organizations and stakeholders. Profoundly, the interface helps organizations to eliminate the loss of information and leverage the data in BIM models to improve communication and collaboration

between architects, engineers, contractors, and facility managers. Recently, a definition of the interface has been adopted in a broader scope to include all common boundaries and non-physical interaction between systems, organizations, stakeholders, project phases and scopes, and construction elements [27]. The interface is a virtual entity whose aim is to help the organization by eliminating information loss. In addition, it improves communication and collaboration between various stakeholders, such as architects, engineers, contractors, and facility managers. Hence, data are supposed to seamlessly transfer through the interface domain between designs, construction, operations, and maintenance [28]. Table 1 details the features of the various interface types.

**Table 1.** Characteristics of the interface types.

| Reference | Interface Type | Details |
|---|---|---|
| [29] | Physical | Physical connections between two or more elements of the building or components. |
| | Contractual | Work packages associated with specialist contractors. |
| | Organizational | Lifecycle relationship between parties involved in the project. |
| [30] | Intrinsic | Physical links among the various components. |
| | Discipline | Theknowledge areas that are necessary to engineer develop studies, analyses, designs, sufficient to utilize the concept. |
| | Project | Strategies among contractors, subcontractors, vendors and any external provider. |
| [31] | Functional | All sub-functions activities and components. |
| | Physical | Interfaces between physical sub-systems. |
| | Hieratical | Between top and low organizational segments regarding project objectives. |

### 3.2. Utilizing Interface

Recording information belonging to managing complaints and responses using emails is not good for solving interface problems. Recently, researchers have developed two critical BIM and interface management (MI) approaches for managing more complex projects [32]. Originally, IM was used as an information-intensive task to provide helpful information to participants [33]. Meanwhile, IM is currently recognized as the most critical organizational strategy in construction management [34]. One of the reasons that made IM an emerging construction strategy was the ability to resolve and enhance construction management by tracking, managing, and eliminating unnecessary mistakes [28]. Hence, project members can locate current interfaces to work out any existing interface issues. It is noted in current construction that without IM implementation, the project could experience design errors, a component malfunction, device performance failures, organized difficulties, and construction disputes [35]. Sacks et al. [14] have emphasized that the BIM models provide a natural interface equipped with sensors and remote FM operations to support monitoring and control practices. Applying BIM in construction management helps project stakeholders monitoring, handling, and tracking all issues relevant to 3D modeling. However, the present BIM-based information systems are still using IM because of the lack of reliable IM communication and management of interfaces (CMI) in the BIM environment [36]. Regarding the history of the construction, the maps of the 3D interface provide information that belongs to the past, present, and future interactions, highlighting an overview of real-time project history.

In construction, interfaces, without distinctive categorization, are either internal (emphasis on contractual relationships) or external (contracts or scopes of work). Based on this characterization, internal interfaces are easier to handle, because they deal with only one team rather than two or more teams, as in the external interfaces. However, when the number of contractors is large, managing interfaces becomes very difficult, and, as such,

it is important to conduct certain classifications between contractors and subcontractors based on the given responsibility [37].

### 3.3. Interface Management (IM)

In the late 1960s and early 1970s, interface management (IM) was introduced to ensure matching the specification of the interface system, data, and missing equipment [38]. Later, IM was used to identify organizational, managerial, and technological interfaces throughout establishing interrelationships [38]. To the best of the authors' knowledge, IM was not wholly integrated into engineering and construction procedures, because of lacking technical infrastructure. In achieving such a goal, several organizations have established IM groups inside their management practices, playing an essential role in training employees to better understand the role of interface manager and interface coordinator.

Interfaces emerged in splitting a project into many sub-projects carried out by several soft or hard, external or internal entities [39]. The soft interface can exchange design criteria, clearance requirements, or utility requirements between the engineering and delivery team and an external party. On the other side, complex interfaces that deal with physical connections between two or more components or systems are examples of hard interfaces. These interfaces include structural steel connectors, pipe terminations, and cable connectors. The interface management process aims to enter into agreements with other stakeholders about roles and duties, time to provide interface information, and early identification of primary interfaces [40]. Having a defined method for exchanging information enables detailed monitoring of performance in meeting requirements, with any inadequacies identified and remedied quickly. Furthermore, interfaces are classified into several groups to fulfill specific goals, such as organizational split interface [29] or resource interface [41].

Figure 2 contains the four main components of the IM system, including the interface of stakeholders, interface points (IPs), interface agreements (IAs), and interface agreement deliverables (IADs). Meanwhile, the interface action item (IAI) consists of tasks and activities aiming to facilitate the agreement of the four IA components between stakeholders. The task and activities are deliverable by IAI to perform an interface agreement (IA), which combines all activities, such as scheduling, drawings, quotations, and evaluation.

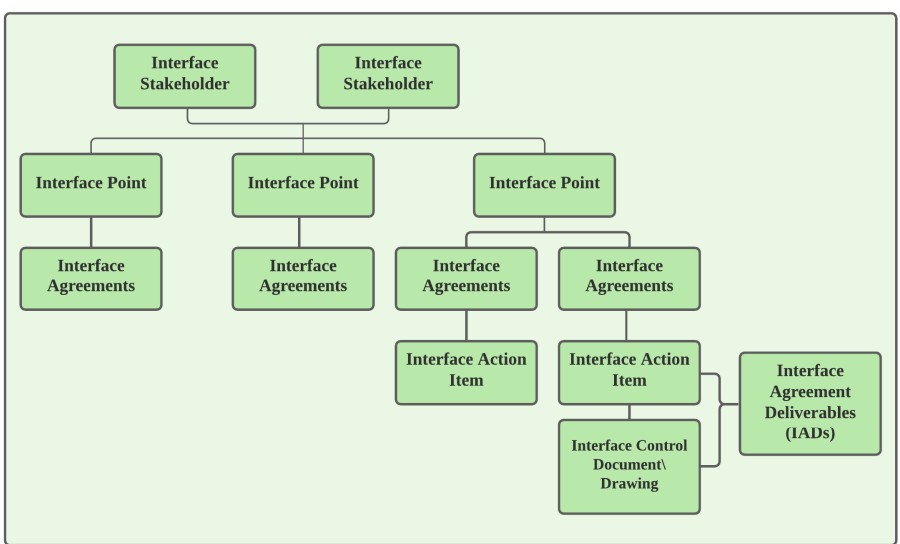

**Figure 2.** Components interface management [27].

### 3.4. Communication and Management Interface (CMI)

The format limitation of the standard BIM file-based model could be used to share the most recent building progress [41]. CMI was integrated into BIM to facilitate discussing, sharing, and responding to issues related to the BIM elemental interface during the

construction phase [28]. CMI enables project engineers and managers to access previous records regarding BIM models for a given project. In the future, it will manage the response to interface problems, as illustrated in Figure 3. The literature focuses on CMI integration; however, it lacks a suitable platform for BIM-based CMI [42].

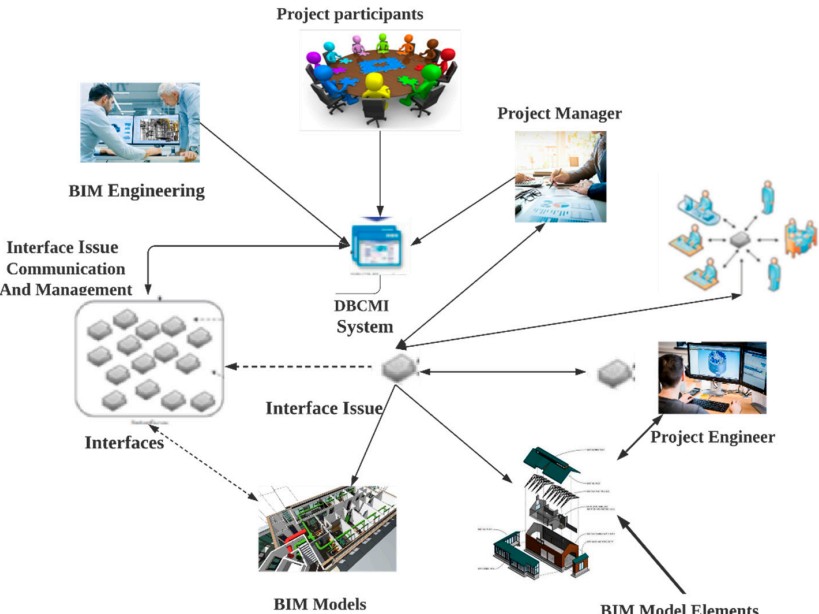

**Figure 3.** Application of integrated CMI in BIM for construction interface management [28].

*3.5. Interface Management System (IMS)*

In 2014, IMS was defined, within the guidelines of interface and IM, as a combination of managerial and relational communication that can be delivered among two or more interface stakeholders. [43]. It was mentioned earlier that IPs, IAs, and IADs are the elements of IMS. IMS was studied in terms of a six-step framework execution, as shown in the self-explanatory Figure 4.

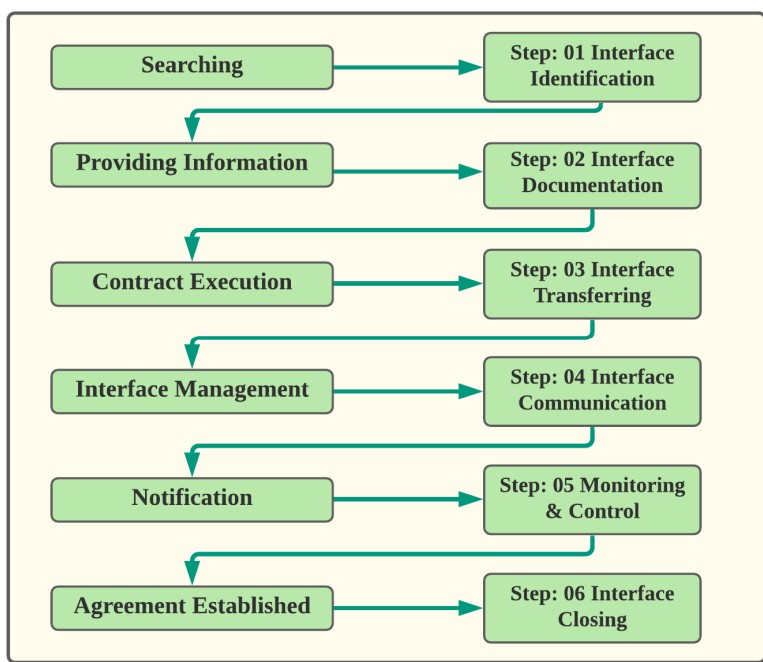

**Figure 4.** Mechanism of IMS framework from searching to the end of the contract.

Recently, there has been increasing concern about IM practices among contractors. Based on the IM definition, a new framework could be used to define the interface management system (IMS). IMS includes many IPs, with each IP including multiple IAs, and each IA may include various IADs [27]. There are several types of interface management, as explained in Table 2.

**Table 2.** Characteristics of the IM interfaces [25].

| Category | Definition/Purpose |
| --- | --- |
| Interface Management (IM) | Managing relational communications between more than one stakeholders. |
| Interface Stakeholder | It is a part of formal interface management agreement of the project. |
| Interface/Interface Point (IP) | It is the soft (hard) contact point between two interdependent interface stakeholders. |
| Interface Agreement (IA) | The formal communication documents between two interface stakeholders concerning desription, actions involved, and dates. |
| Interface Action Items (IAI) | IAI regulates tasks and activities to perform the defined agreement in each interface agreement. |
| Interface Control Document Drawing (ICD) | To identify and capture interface information prior to approvement. ICDs are useful for separate organizations with a common particular interface. |

Based on the above discussion, interface stakeholders are involved in many deliverable information or tasks to handle the interface efficiently. In each interface point, there are numerous interface agreements. These agreements can be delivered to other parties. Each interface stakeholder can deal with several interface points and agreements, as illustrated in Figure 5.

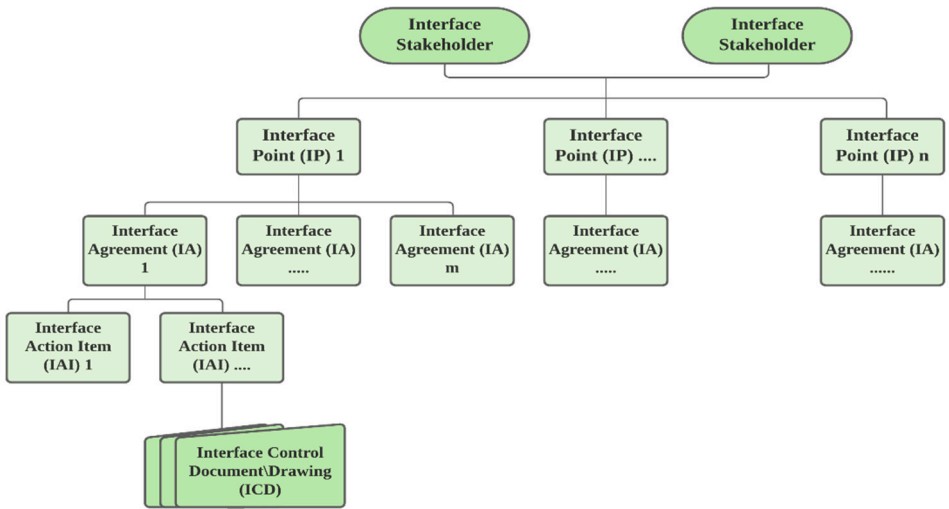

**Figure 5.** Hierarchy of Interface Management Elements [25].

### 3.6. Application Program Interface (API)

API was defined in 1968 as a collection of code routines to provide external users with data and data functionality that was used in programming libraries [44]. API was also used interchangeably with frameworks, libraries, and operating systems. Nevertheless, the current colloquial definition of API refers most typically to a synonym for web API [45]. According to Programmable Web-based API directory, the number of available APIs continues to expand, particularly those classified as data, financial, or analytics [5]. Numerous accessi-

ble APIs enable access to massive data volumes. APIs are versatile technical solutions that may be utilized in various applications. The first application is the Google Maps Platform, which has a Places API that provides access to over 150 million locations worldwide. Firms use APIs to refer to products, add more data to databases, or create specific APIs [46]. The second application is to perform functions related to procedural languages, such as C, to act as a function call, by involving information about all the functions and routines that it provides.

APIs are a collection of methods that enable programs to access data and communicate with external software components, operating systems, or microservices. APIs are a critical component of many modern software architectures, because they provide high-level abstractions that simplify programming processes, create distributed and modular software systems, and allow code reuse [47]. Hence, APIs make necessary accessible functionalities for developers to enable IoT cloud infrastructures [48]. APIs are digital apps that can help in communicating with back-end services [45].

AP can be expanded to describe all calls, subroutines, or software, to enable application programs in services such as application, operating system, network, or another lower-level software program [49]. APIs facilitate information for developers to work with essential capabilities or data to leverage and govern IoT cloud infrastructures. APIs allow partners or the public to activate participants and generate new revenue streams [50].

Additionally, APIs help establish an interface that connects functions in one unique system, cutting down transaction costs, and improving efficiency [51]. Another feature of API is providing third-party developers with access to private data owned by Google, Facebook, Twitter, and many other large firms [52].

APIs, then, constitute the interfaces of the various building blocks that a developer needs to create an application [53]. An API currently summarizes a set of programming codes to transfer data between one software product and another. APIs are composed of two fundamental components: technical specifications describing data exchange choices between solutions in the form of a request for processing, and data delivery protocols and a program interface based on the specifications they represent. Today, APIs come in three types—standard, widespread, and versatile. Figure 6 explains the main types of API and the corresponding policies.

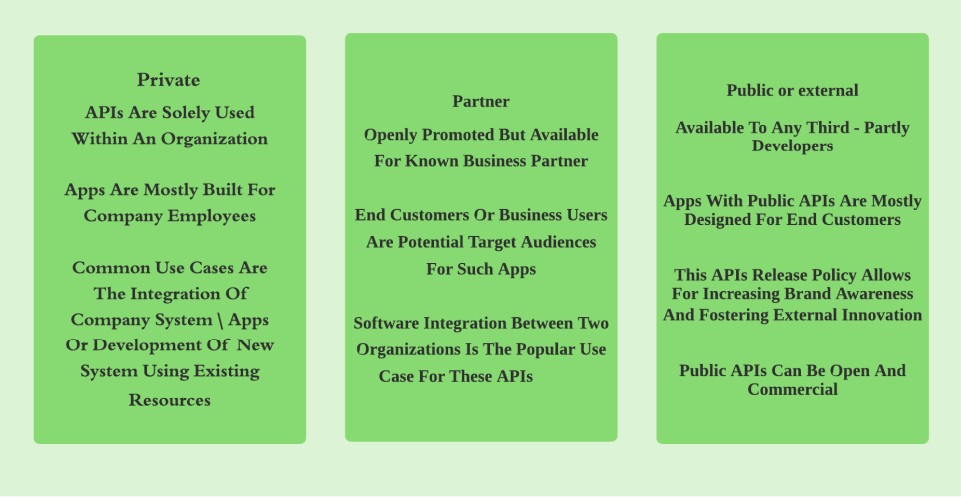

**Figure 6.** API types and functions [51].

## 4. Industry Foundation Class (IFC)

Sheng Jun et al. [54] proposed several methods to transfer various software programs and data formats using IFC or DWG. There are many approaches to co-ordinate IFC, IDM, Open BIM Collaboration Format (BCF), Open BIM Collaboration Format (BCF), and Model View Definition (MVD). For instance, the formats of BIM can open communication between

two users based on the IFC standard data model. IFC is equipped with a high degree of interoperability that can facilitate the opening data standard known as buildingSMART (ISO 16739), depicting the whole building geometry. In addition, IFC provides digital building models to help architects sharing the BIM environment. The Information Delivery Manual (IDM) is a systematic tool for identifying and specifying information flow during a facility's lifecycle. Furthermore, IFC has been developed by buildingSMART and registered under ISO 29481-1:2010 and ISO 29481-2:2010. Concerning IFC schema needed to satisfy one or many ERs, MVD defines a subset published by the software tool ifcDoc developed by BuildingSMART [55]. Furthermore, Zhang [55] considers BCF is an important tool to exchange information in terms of queries, ideas, or demands between different software products, resulting in a technological solution for communication among stakeholders.

In addition to addressing the IFC data model, BCF addresses the position as defined in snapshots or camera perspectives. It is well-known that transforming data into another using software applications with heterogeneous models can be conducted with a single multimodal [7]. Hence, the common practice of BIM models is to exchange information about building structures throughout their life cycle, which is a standard industry practice. The IFC standard in BIM applications acts as a medium for data exchange across domains and stages [56]. The domain of exchanging data for BIM modeling resembles the IFC scheme subset [57]. The benefit of IFC-mapped data exchange is to help the software vendors developing practical import/export features to allow project participants to share and exchange BIM model information. During the different stages of projects, BIM plays an important role in exchanging data and information with specific formats amongst architects, engineers, clients, and contractors to serve throughout the project lifecycle [58].

In construction, stakeholders rely on each other to acquire details. The most critical issue here is to automatically interpret and process the information mapping into data for BIM applications with cross-domain and inter-stage coordination [56]. This process leads to an automated system that needs reliable interoperability for marketing and technological levels [59]. The interoperability process requires exchange information for all contributors to understand the need and provide this information for usage. The goal of interoperability is to provide a better communication system that can be placed at various levels and contribute to achieving the result. Interoperability creates the significant digitalization of the whole process towards full automation and efficient management of these processes [8]. The preparation to establish a construction is a multi-facet issue that started before the initiation of the construction and continued for the whole lifecycle.

One of the most important matters is the scope of each construction, which consumes time with the collaborators throughout the project phases. In this case, there is software to be developed by architects, structural engineers, and designers to store and analyze data. This is called heterogeneous information, since the data are stored, shared, and preserved in different realms to ensure consistency [60].

The construction industry domains involve distinctive advanced data exchange for BIM models, using specialized fields of architecture and construction such as neutral formats found in the industry corporation categories. The IFC schema for diverse disciplines should define the type of BIM standards [61]. IFC, an exchangeable neutral format, is often used in design, engineering, manufacturing, and facility management [62]. The data exchange amongst various software across BIM models and the relevant incremental or "as-built" collection archiving is the main scope of the IFC applications [61].

## 5. Developing Framework

The framework is defined in different ways, and for various purposes. In the early 1980s, Model-View-Controller (MVC) was the first object-oriented framework [63]. Since then, several papers have been published to show the broad and spread nature of using the term 'framework'. The most reliable definition for the framework is that the one connected to software engineering, which refers to designing and implementing large object-oriented software [4].

However, in a different approach, the framework known as conceptual structure is heavily used to solve or address complex issues containing tools, materials, or components [64]. The framework could be defined as an object-oriented form that can be embodied classes and yielding a solution to a family of problems [65].

Technically, the framework represents a reusable design for a system wholly or partially by setting the abstract classes and their interaction [66]. Another standard definition for the framework depends on formulating a skeleton of a customized application to be suitable to an application developer [67]. Despite the several definitions due to the framework's purpose and nature, the general definition of the framework could include structure, aims, and interrelated parameters within a particular phenomenon. Hence, the framework is a comprehensive architecture that outlines the decomposition of a program into a collection of interacting elements [68].

### 5.1. Framework Applications

Based on multiple definitions of the framework, it can be helpful for various purposes. Framework, as an application, has become a set of elements for designing or developing a reusable code. The technical difference between a framework and an API is that an API is only a part of a framework. The development framework is being employed to present data integration climate, as corporations seek to decrease cost by outsourcing project-based solutions to temporary staff and third-party firms. Wu and Simmons [69] have confirmed that project planning is vital in the current software development process. Hence, it is necessary to aid in the comprehension and application of the answer [70].

There are two types of frameworks: theoretical and conceptual. This study distinguishes between these two types of the framework, since those two types provide direction and stimulus to study and extend knowledge. According to Grant and Osanloo [71], a theoretical framework is the 'blueprint' or guide for research or providing a specific theory or set of theories concerning a particular area of human effort that may be used to analyze occurrences. By using the theoretical framework, research endeavors will get several benefits. Researchers show how they define the study philosophically, epistemologically, methodologically, and analytically with the addition of the structure [71]. The theoretical framework serves as a guide, and should be consistent with all aspects of the research process, including the formulation of the problem, the review of the literature, the methodology, the presentation, and discussion of the findings, as well as the conclusions derived from them [72].

### 5.2. Framework for Energy Management

The energy sectors in the construction are a comprehensive and essential part of the construction, which is denoted by the Building Energy Management System (BEMS). Hence, it is expected that integrating BEMS in BIM creates an effective energy data monitoring framework using the human–machine interface (HMI) [73]. Researchers investigated the related technology trends and derived BIM-based HMI framework requirements by identifying the role of each component of the framework. Furthermore, an interface is designed between BIM and BEMS with consideration of HMI, and a well-prepared questionnaire.

According to Siao et al. [74], IM has been recognized as the most critical organizational strategy in construction management, because IM is fundamentally reported as a routine process operation guided by specific control of communications [75]. The need for IM and IMS has become more apparent as the construction becomes more complex, with a considerable increase of participants [41]. The gap inherited from several academic studies was the lack of systematic approaches for managing interfaces during construction and assembly phases [76]. In 2009, Lin [24] had identified four primary interface problems, including insufficient platforms for construction project management, improper managing interface conflicts, problems of managing time, space, and efficiency during the construction phase, lack of an effective mechanism for tracking and managing interfaces, absence of

complete official record amongst participants, and difficulty of tracking interface events and obtaining interface information from other participants [33].

The role of IM in tracking all participants' involvement could lead to improving operational management, minimizing detrimental change, and enhancing beneficial change. Morris [38] identified two interfaces—static and dynamic. Furthermore, other interfaces, such as personal, organizational, and system interfaces were identified by [39]. More closely, Pavitt and Gibb [29] proposed three main interface types: physical, contractual, and organizational. The significant number of interfaces could create difficulties in applications. Hence, to simplify this issue, the search for an interface should be aligned with the construction phase. In this phase, interface problems can be categorized as construction, processing, space-related, communication, and variability problems interfaces [33].

The development of 2D to 3D patterns improves the shape, size of a component, and spatial relationships between the components. BIM, as a digital tool, can continue updating and sharing project design information [77]. However, the 3D pattern requires precise geometry to support the design, procurement, fabrication, and construction activities [9]. Accordingly, BIM-based visualization could express information more intuitively by realizing real-time construction [78]. In addition, 3D also provides participants mindful of accuracy and adequacy [79]. BIM and CAD share similar views concerning the construction interface management and develop ConBIMIM system, a mixture of construction, BIM, and IM [33].

BIM is a comprehensive system which enables participants to track project updates and to proide data and information about models whose aim is to manage the effects of the databases on a specific model, capturing information from a particular model, and preserving adding industry-specific applications [80]. IMS, on the other hand, is the source for providing a simple and straightforward representation of various interfaces; clarifying the events of the current interfaces; extending the relationships among interface events, and helpings BIM users to track and identify interface events using different colors [33], as shown in Figure 7.

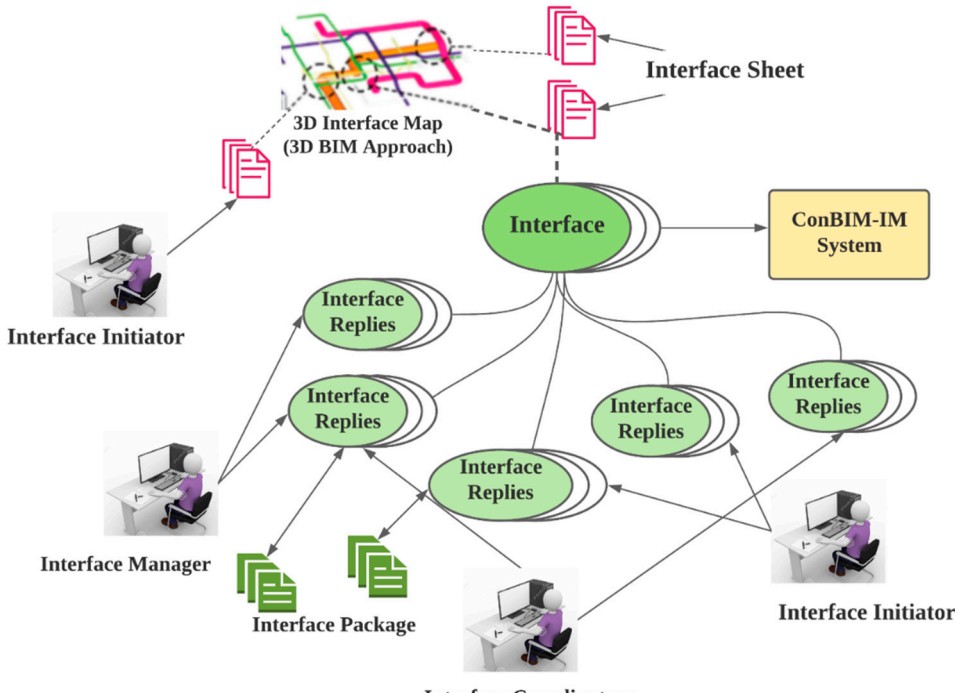

**Figure 7.** The mechanism of interface management system [33].

The ConBIM-IM system was proposed to design by constructing IM and IMS. Meanwhile, the 3D-CAD interface represents objects and attributes of interface events—the BIM

stores digital interface information to facilitate easy updates and interface transferring. As a result, the 3D interface information can be identified, tracked, managed, and further solved problems. The ConBIM-IM enables participating engineers to share and save all documents in 3D formats, and be available upon future request. Figure 8 details the 3D interface maps framework equipped with the eight components of ID, topic, date, description, owner, people, attachments, and history.

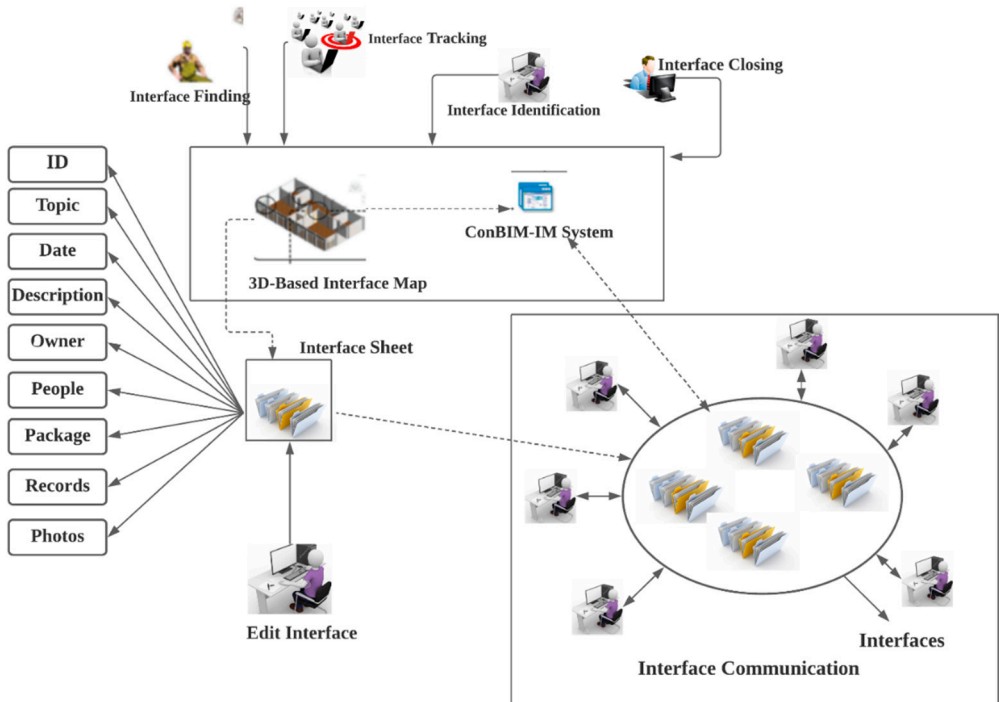

**Figure 8.** The concept and framework of the 3D-based interface maps approach [33].

*5.3. A Typical Framework*

Figure 9 shows the BIM-based interfaces framework communication and management integrated with the Interface Breakdown Structure (IBS) and MBS. The process then creates an IBS and a Model Breakdown Structure (MBS) before integrating them in BIM. IBS can break IM into elements of related interfaces. Meanwhile, IBS is a hierarchical representation of interfaces, starting at higher levels, and increasing to more acceptable level interfaces. Furthermore, MBS in interface management is a deliverable-oriented breakdown of a BIM model into more minor elements for interface management. MBS is a crucial interface integrated with elements of BIM models. The CMI-related information stored in elements of the BIM model includes both CMI-related problems and solutions [81]. The CMI essential information should include the interface description, responding, or related attachments such as documents, reports, drawings, and photographs. CMI then enables communication and activates responses associated with projects, activities, people, and organizations. Identifying the connection between the information of CMI and the corresponding interfaces is crucial to the project's management.

In addition to these developments, project engineers can acquire CMI-related issues before sharing them with corresponding BIM model elements. The 3D BIM model known as the DBCMI system can be illustrated at different CMI access levels depending on user roles. As the information is updated in the DBCMI system, the server automatically informs corresponding participants by sending e-mails to the project participants. CMI is equipped with an initial stage through which all responsible participants or project managers are identified. The second stage allows the project participants to edit the information sent and select the appropriate BIM model. In the final stage, the engineers can submit the interface issues associated with the BIM model elements to the DBCMI system for approval. After

the approval stage, the corresponding participants respond to problems via the selected interface in the DBCMI system. The system can track all these activities to show the status and the results for each interface problem [81].

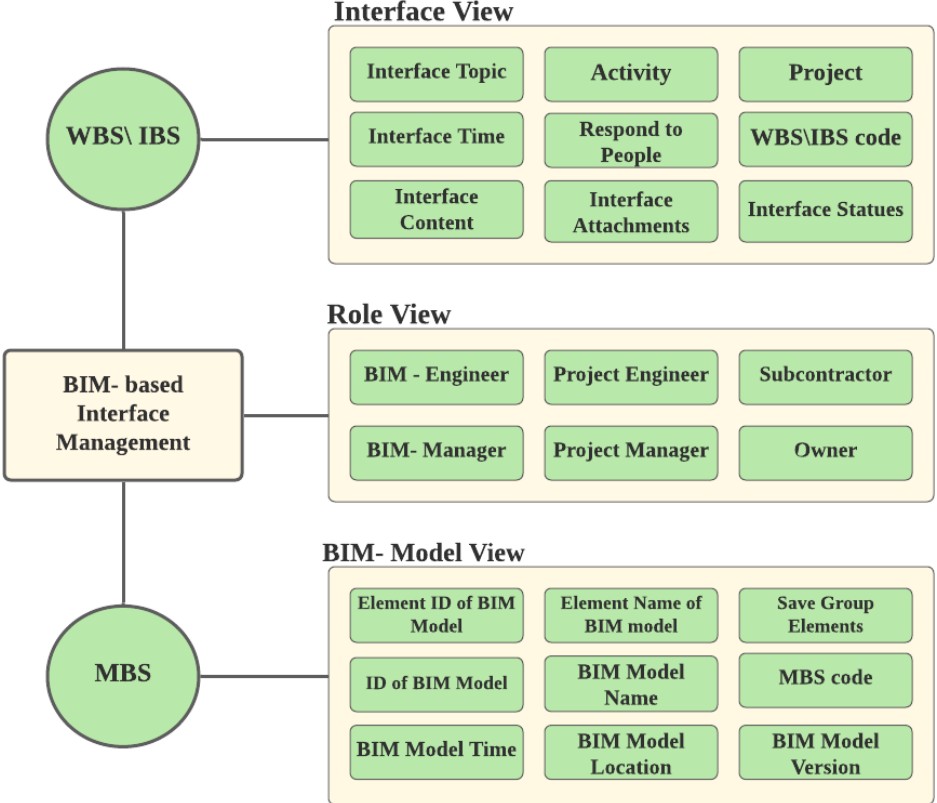

**Figure 9.** The framework of database- and BIM-based interfaces communication and management [28].

There is another approach to generate a dynamic energy simulation model for a single existing building by collecting existing data to prepare energy retrofits at the lowest cost possible. The proposal includes establishing a polygon model by employing photogrammetrically generated point clouds with the Tool for Energy Analysis and Simulation for Efficient Retrofit (TEASER) and AixLib. A single-family house was taken as a case study to achieve the purpose. The model reproduces the internal air temperatures during synthetical heating up and cooling down, with building heat transfer coefficients (HTC) agreeing within a 12% range. The model requires accurate window characterizations and justifies the use of a very simplified interior geometry. However, uncertainties arose regarding comparing different typologies showing differences in pre-retrofit heat demand of about $\pm 20\%$ to the average [82].

In modern environments, high-rise buildings have become indicative of a diverse building environment that requires special treatment by monitoring activities such as fire hazards. BIM limits fire accidents by creating, developing, and implementing an integrated fire disaster prevention system. The disaster response system is composed of a complete plan, including prevention and evacuation. However, this alarming disaster system is prone to human errors, wrong location, poor communication, and incompleteness. However, the role of BIM is to minimize possible human errors. The system could be better performed in case of providing 3D visualization to support the assessment, planning, and detection of fire safety [83].

## 6. Review Previous Methodologies

### 6.1. Integrating CMI

Lin et al. [28] conducted a pilot study to investigate the connection between CMI and MI by interviewing project managers and expert engineers. The study concluded that there was no suitable platform supporting IM, incomplete records for communication and supporting documents, and no clear distinction for how these problems were developed. Hence, CMI must be fully incorporated in construction processes and activities to satisfy these problems, recording all communications at each interface, and linking CMI to BIM models. Lin et al. [28] proposed a concept in which IM was prepared for a general contractor (GC), as shown in Figure 10.

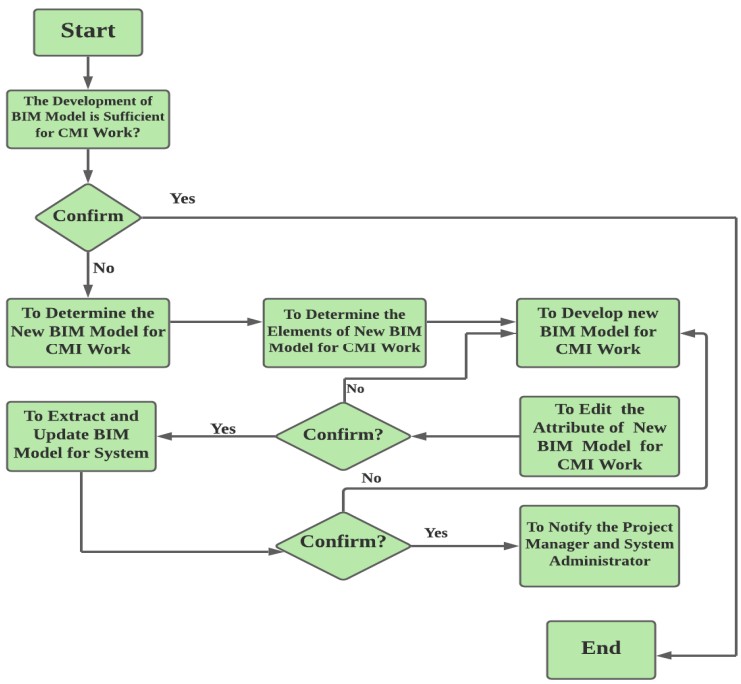

**Figure 10.** Investigation of CMI and BIM model [28].

### 6.2. Integrating IM

IM integrating into BIM can be considered a robust approach to improving project monitoring and control to enable real-time decision-making. Most constructions are currently using IM and BIM separately, and, hence, combing IM and BIM is very useful for management, especially for deterministic product management perspective and to a better understanding of managing the complexities, uncertainties, and risk in organizational structure, coordination, collaboration, and communication [32].

The complex construction projects require creating a BIM model before establishing its IM system using a conceptual BIM model that can be generated and detailed during the project lifecycle. On the contrary, an IM system starts in the design phase as part of the project's dynamic systems packages that include changing or evolving elements or removing items from the system. In the construction design phases, many new elements may suffer from repeating, cancellation, or modifying. In all such cases, editing the project essentials has become a necessary step to follow-up the execution of the project. These changes may require updating the project participants as IPs change on the IM system during the project lifecycle. As the project moves from a particular phase to another, the number of participants increases. Then, the number decreases towards the end of the project. This behavior means that IM expands and shrinks, with the change in the number of project participants affecting the number of interface points during the project lifecycle [32].

The IM system consists of interface points (IPs), interface agreements (IAs), and interface agreement deliverables (IADs), as shown in Figure 11. In addition, IPs may contain many IAs, with each IA possibly including many IADs, which means that a typical project may include tens of thousands of IADs which need accurate management to design phases of complex projects that work towards reducing cost and improving [25].

Since 2013, researchers have been steadily developing an effective web-based IM system platform, such as Lin [84], who instrumented a connection amongst project participants for managing interface problems during the construction phase. Within four years, Ju et al. [85] had successfully developed an integrated interface model through which the traditional methods have been changed to a more standardized and structured aiming at improving an IM system. During the period from 2013 until very recently, the heavy research on IM was not successful in eliminating the gaps about visualizing the IM system in the design phase [86].

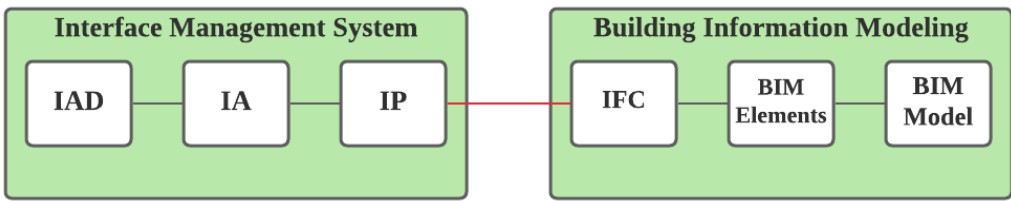

**Figure 11.** The mechanism of connection IM and BIM [27].

### 6.3. Document Management System

Complete knowledge about the building material and corresponding repair and condition information is provided in the 'SharePoint' electronic document management system (EDMS). The most crucial benefit of EDMS is that storing information can be done without a formal structure. The information becomes available and can be obtained at any time to be used as needed. Recently, the EDMS information has been stored with some structure to the documents to transfer information quickly. The requirement of the information could become a computable BIM data parameter and to enable reviewing the existing template documents [87].

Regarding the Asset Lifecycle Information Management (ALIM), BIM could benefit from the open standards that enable future builders and developers to record information on the entire lifecycle. ALIM arises because of the availability of information as an essential part of structural digital twins. This type of system provides detailed insights into performance and processes through simulation. This information can be combined and reused quickly if specified, designed, realized, managed, maintained, or dismantled according to open standards. Linking and connecting data are innovative concepts, since ALIM requires bringing together diverse sources from different domains with their standards; knowledge of various standards is essential. Therefore, ALIM provides extensive expertise on both data integration and data modelling based on linked data and (inter)national open standards [88].

## 7. BIM-Based Algorithm

The algorithm describes optimization problems with different variables confined by existing conditions and constraints [89]. Solving complex optimization problems in engineering, economy, and science requires utilizing metaheuristic algorithms to find solutions within a reasonable amount of time. Optimization problems are nonlinear, multimodal, and generally subjected to a set of complicated constraints. One of these constraints is the existence of different and conflicting objectives of a single problem which make finding an optimal result difficult [90]. BIM has been considered a comprehensive model for optimizing different construction processes in all stages before and after the project [91]. BIM could provide a suitable framework that supports the decision-making process by utilizing all necessary information at the right time.

Algorithm optimization has been heavily utilized as a decision-making process in construction in the planning of site layout, because such an optimization results in enhancing productivity by facilitating the movement of labor and materials. In addition to saving time and cost, BIM can assist in planning the construction site layout [92]. Implementing this process is not easy, because it relies on many interlocked factors that influence the site layout problem [93]. Hence, it is imperative to create a suitable algorithm to achieve the highest possible case [94]. It has been reported that the optimization process is a complex method, due to the limited number of feasible configurations that lead to obtaining exact methods [95].

The solution seemingly relies on metaheuristic algorithms defined by [96] as a framework of a highly independent algorithm that provides guidelines and strategies for utilizing the optimization process. These algorithms show high compatibility with many engineering optimization problems [97]. Metaheuristic appears in different forms and for various tasks. The main classification of metaheuristic algorithms is either trajectory-based or population-based algorithms. Away from the complications of these algorithms, it can be said that the global utilization of the search algorithms shares the same or very close purposes [98].

### 7.1. Algorithm Roles in BIM

In recent years, studies have been trying to implement metaheuristic algorithms in BIM-site layouts. As examples, genetic algorithm (GA) and particle swarm optimization (PSO) were among metaheuristic algorithms. The applications targeted the period during planning to support the decision-making process for construction projects [95]. Achieving this algorithm application implies that the site layout is considered in two fashions: a static and a dynamic model. The static model facilitates the initial plans until the end of the entire construction phases [99]. On the other side, dynamic layout models identify the required duration of each facility [100]. The input data and corresponding constraints that characterize the optimal layout problem were discussed by [92]. The optimization process can be mainly achieved by considering the dynamic models, which supposedly contain any possible change throughout the construction phases. In the optimization process, the mathematical procedure may consider integrating generic algorithms to better facilitate the use of a radio frequency identification (RFID) system that depicts tracking object location in the real-time procedure [101]. For the materials, [102] employed a generic algorithm for dynamic plan optimization while using a metaheuristic algorithm to optimize both the material and personal movements. Furthermore, the A* algorithm is used to find the shortest distance between multiple points. Use of the A* algorithm is to counterpart some obstacles in the construction site [103].

Another benefit of using optimization is reducing the time required of travel frequencies at various construction phases [104]. The other application targets other applications in construction projects using integrating BIM product models with several algorithms to achieve optimization, which should be conducted in the auto-generated schedule [105]. As shown later, the BIM simulation system uses a 4D model and generic algorithms to obtain an optimal construction schedule [106]. Furthermore, BIM was utilized to develop and generate construction schedules [107]. These approaches require efficient algorithms to achieve a very high computational speed for optimization using hardware with field-programmable gate arrays [108]. The optimization for maintaining the life cycle cost throughout saving energy was studied by [109].

Sustainability was also studied by integrating the BIM model and the famous multi-objective particle swarm optimization (MOPSO) [110]. BIM is used to reduce the computational time in building fire emergency response operations using the metaheuristic algorithms [108].

### 7.2. K-Means Algorithm

The tools assessment of the BIM performance focuses on qualitative aspects, such as the success of BIM project progress or the capability of the construction company to complete a project [111]. However, the most challenging part of this type of qualitative assessment is outlining the qualitative BIM performance assessment tools for the operational strategy. Kim et al. [112] have solved this matter by proposing a method based on the k-means clustering algorithm. The solution implies that the BIM performance assessment system enables both the evaluation of the current BIM execution ability and the corresponding prediction of the cost-effectiveness before and after implementation of BIM projects through a comparison with other projects. The method is based on the expectation of a k-means clustering algorithm by analyzing the return-on-investment (RoI) approach.

Previous attempts to solve the same problem relied on various approaches such as implementing BIM initiatives [113], analyzing the BIM substantial benefits and characteristics in estimating the cost-effectiveness [114], analyzing the 13 risk factors (technology, human, management finance, and others) required to consider a swift counterstrategy that results in the success of BIM project or developing six performance indicators of the quality control, schedule conformance, total cost, unit particulars, cost per unit, and safety [115].

Searching for similar project groups can be facilitated using the k-means algorithm, since this algorithm is very efficient in clustering analysis, which quickly results in stable results [112]. A procedure runs the processes associated with the k-means algorithm. The procedure is comprehensive, as it is characterized by input information, metrics to weigh and calculate similarities, and searching for similar projects. The data used for these clusters can be determined by the similarity ratio between the case and target projects, as depicted in Table 3. The metrics are used in evaluating project information and assessing BIM tool, BIM application phase, performance capability, costs, and usage frequency [112].

**Table 3.** Metrics for analyzing the BIM environment [111].

| Metrics | Description |
|---|---|
| Project information | Project name, type, cost, country, region, etc. |
| Goal of BIM introduction | Producticvity improvement and schduling reduction in cost. |
| BIM tool | Revit architecture, Bentley architecture, ArchiCAD, Navisworks, Vico control, etc. |
| BIM application phase | Design phase-conceptual, construction phase, etc. |
| Performance capability maturity | Organization, technology, management |

### 7.3. Heuristic Optimisation

A heuristic (a self-discovery) algorithm is a shortcut that allows people to solve problems and make judgments quickly and efficiently, and to find a near-optimal solution. This application means that heuristic algorithms shorten the time to decide, while allowing people to function without constantly stopping to think about their next course of action. This result was subjected to a trade-off balance between several factors, including optimization, accuracy, preciseness, completeness, and solution speed. Heuristic methods were typically employed when the classical solution failed to achieve an exact solution. For large datasets, users must define the objectives to optimize an algorithm using heuristic methods [116]. Despite the potential challenges that visual programming languages (VPLs) pose, they are still offering several benefits. These benefits range from easily creating designing industrial programs with minimal computer science training, to easily accessing APIs platform. Besides, the design is characterized by simple geometry, flexibility, and easy integration that support the automated analysis of non-parametric features [117]. Figure 12 shows the general process for computational algorithm development, which includes various computational procedures, such as Boolean, vectors, and, most importantly, heuristic

optimization methods [118]. However, BIM computational processes require many more processes and analyses; hence Seghier et al. [119] proposed different quantitative and qualitative processes which can be computed using algorithms. This type of data analysis forms a specific data management platform that can handle algorithms.

The application of the VLP-algorithm for building design is conventionally practiced according to the workflow presented in Figure 13. Hence, the design can be checked manually by engaging NLP with the auto computational performance. The workflow outlines the initial steps of the BIM model, the logical statement inputs from NLP-based analysis, and the automated compliance checking module.

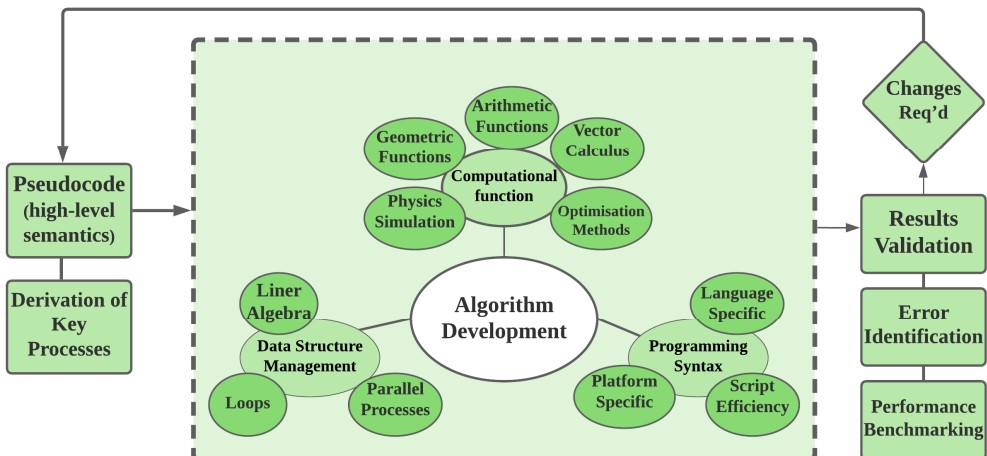

**Figure 12.** General process for the development of VPL-based computational algorithms for geometric optimization in BIM [112].

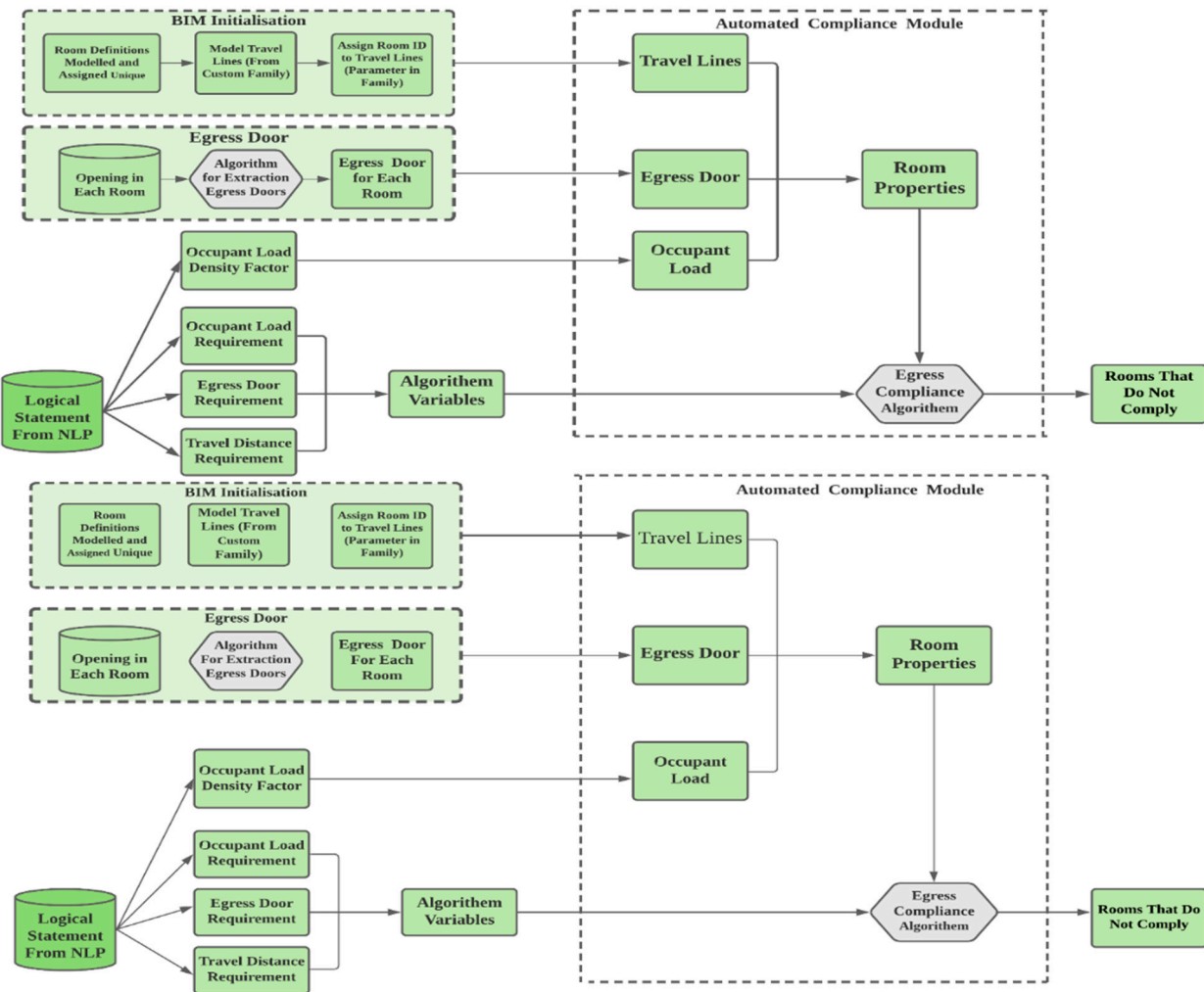

**Figure 13.** The compliance of the automated and logical building codes using NLP [112].

## 8. Summary of Previous Empirical Research

Based on the paper title, the role of the interface and interface management, the concept of the multi-model and the relevant applications, and the role of algorithm optimization are highlighted in Table 4. The table shows five elements: objective, methodology, gap(s) presented, and the contribution of each article. One of the most proper methodologies to reduce BIM complexity is the automation, as Mukkavaara [120] reported, who proposed BIM-based automation in the design process. The reason for automation is to manage well in every single application of BIM. This issue was discussed from a different point of view by integrating interface management building with BIM, as comprehensively discussed by Eray et al. [32], aiming at developing a framework that can be used as a good coordinator for communication amongst participants over interface-related problems in the project definition and design phases. The reason for this interface integrating is the ever-increasing complexity of constructing projects, such as power plants and rapid transit systems. Meanwhile, the concept of BIM interface was further discussed by Lin et al. [28] due to the integrating of BIM and web technology, aiming at allowing the user to communicate and explore the various links of BIM.

The main purpose of creating a BIM interface is to allow general contractors to enhance their CMI work efficiency during the construction phase. The BIM interface was introduced by Kang [73] under a different form of human–machine interface (HMI) to monitor the energy management system regarding energy consumption. The results of introducing such as a measure were positive for both the effect and the benefits. Tang et al. [58] have proposed

a methodology that targeted smart buildings using a building automation system (BAS). BAS can fill the gap caused by the difficulty of performing the exchange of information amongst BIM stages. This development depends on introducing the network system in IFC by mobilizing the interface. The implementation of BIM faces some barriers in ACE, as reported by Leśniak et al. [121]. They proposed a technique known as the Ishikawa procedure, in which education, including training and studying BIM technology, could result in a better understanding of MIM implementation. The computational BIM is a vast field utilized to automate BIM by optimization to achieve higher efficiency in critical fields, such as better building materials, opening sizes, and glazing types (Lim et al. [122]). The optimization was carried out to serve to integrate Revit tools, dynamo visual programming tools, and multi-objective. As a result, Lim et al. [122] have contributed a series of tools integration using MATLAB to facilitate the possibility of automating and speeding up the process of retrofitting constructions.

The idea of exploring the multimodel approach in BIM was enhanced by Pruvost et al. [6] in projecting uncertainties in designing space by collaborating with the building design workflow. As a result, they integrated several disciplines to share information from different data models and formats, to eventually be used as input in building energy analysis (BEA), including geometry, energy infrastructure, weather, and building usage. Another research study accompanied the progress made by Pruvost et al. [6]. This was in the same field conducted by Fuchs and Scherer [7], who approached BIM multi-model throughout nD-modelling, based on the available original data.

BIM is involved in structural sustainability, as part of the effort to treat the environment better, as reported by Oti et al. [123], who have utilized API in the extended demonstration of the conceptual design option of BIM. This demonstration was conducted by modeling and creating algorithms able to enhance nD building performance. The gap that prompted using nD building was to exploit expanding BIM scope. Another trial to enhance the performance of the multi-model was experienced by Cheng and Cheng [124], in which a genetic algorithm (GA) was employed for better natural selection. The enhancement was carried out by employing Markov chain theory to determine the criteria of adaptive termination with minimal cost. The complexity of MIM applications can be reduced using several techniques.

In 2019, Deng et al. [81] introduced a new parameter in BIM applications as they considered the safety measures a part of the BIM emergency management plan through the Revit platform. In this attempt, Navisworks software was employed. By employing an emergency plan, BIM has become involved in a more detailed approach in construction management. In addition to the work of Pruvost et al. [6] and Lim et al. (2019) in integrating BIM with various techniques as discussed earlier, Li et al. [125] prosed another way to integrate web services with BIM to improve the early design processes called Dynamo BIM, while [126] investigated some approaches to integrate BIM, IoT, and FM for renovating existing buildings. All integration trials mentioned above were validated and then assessed in estimating energy consumption.

The simulation technique is widespread in many fields. In BIM, Siegele et al. [127] used a new MATLAB simulator to study the construction's dynamic feature by programming the object-oriented language of MATLAB. The results have shown that the indoor quality has improved. The last article was chosen to present the optimization needed to upgrade the efficiency of BIM applications in specific fields, such as optimizing the energy consumption and the space occupied by the project.

In this sense, Amiri et al. [94] have suggested a metaheuristic algorithm to support the decision-making process: Uses in planning construction site layout. It has been shown that metaheuristic optimization algorithms have been recognized as very famous hybridizing algorithms that can work very well with the k-means approach.

**Table 4.** Summary of the most relevant articles.

| # | Author/Title | Objective | Methodology | Gap(s) | Contribution |
|---|---|---|---|---|---|
| 1 | Mukkavaara [120]. Structures for supporting BIM-based automation in the design process | To investigate the structures that can be applied to support automation within a BIM-based design process. | Exploring different methods for automation workflows using three studies. | The complexity of the design process s not managed well in each single BIM application. | Providing the foundation for mapping between multiple sets of data to resolve the coupling of information at each activity in an automated BIM-based workflow. |
| 2 | Eray et al. [32]. An overview on integrating interface management and building information management systems | To develop a framework for integrating IM and BIM systems to create better coordination and communication between project participants over interface related problems in the project definition and design phases. | By explaining the relation between IM and BIM systems, a framework was developed to connect interface points within a 3-D model followed by validating the functionality of proposed framework. | The complex projects such rapid transit systems, power plants, refineries and port facilities were facing difficulties in managing execution due to geographical specialized location. | Integrating IM and BIM systems improves better execution process by improving the project control, communications and alignment along with reduced requests for information, change requests, and rework. |
| 3 | Tang et al. [58]. BIM assisted Building Automation System information exchange using BACnet and IFC | To link smart buildings to different building systems together with the Building Automation System (BAS). | Use IDM and MVD methodologies to define an IFC subset schema so that BAS information conforming to the BACnet protocol can be represented in IFC data model for information exchange throughout various project stages with BIM tools. | It is rarely seen to design BAS or exchange BAS information in different project stages using BIM tools. | Facilitate information exchange for BIM-assisted BAS design and operation using one BAS open communication protocol named Building Automation and Control Networks (BACnet) and open BIM standard Industry Foundation Class (IFC). |
| 4 | Lin et al. [28]. Construction database-supported and BIM-based interface communication and management: a pilot project | To integrate BIM and web technology to construct projects allows users to communicate interface issues and obtain responses for them effectively. | Using a case (pilot) study in a building project by proposing database communication and management interface (dBCMI) system. | The absence of suitable and necessary systems or platforms to tackle the communication and interface management. | Developing a database-supported and BIM-based CMI (DBCMI) system for general contractors to enhance their CMI work efficiency during the construction phase. |
| 5 | Kang [73]. BIM-based human–machine interface (HMI) framework for energy management | To introduce Building Information Modeling (BIM)-based Human–Machine Interface (HMI) framework for intuitive space-based energy management. | Introducing BIM-based HMI framework after deriving the considerations and requirements necessary for linking the energy control point and BIM through a questionnaire designed by practitioners. | The absence of effective heterogeneous link between BIM and energy management system to provides space-based real-time energy monitoring. | A positive effect (3.9/5.0) on the connectivity of BIM-based HMI with benefits (4.3/5.0) for real-time data monitoring. |
| 6 | Leśniak et al. [121]. Barriers to BIM Implementation in Architecture, Construction, and Engineering Projects—The Polish Study | To analyze the cause and effect of identified barriers (failure) to implementing BIM technology in the construction process in Poland. | Employing a tool that helps to recognize the actual or potential causes of failure known as Ishikawa. | Limited information about the influence of the poor BIM implementation in Poland and about the awareness of reducing the obstacles of BIM implementation. | Introducing factors that are needed to better implement BIM, such as education, training, and studying BIM technology |
| 7 | Lim et al. [122]. Computational BIM for Green Retrofitting of The Existing Building Envelope | To automate the computational building information modelling (BIM) in decision-making for green retrofitting of the existing building. | Integrating Revit tool, dynamo visual programming tool, and multi-objective optimization algorithm to optimize overall thermal transfer value (OTTV) and construction investment cost. | The need of better and efficient decision to optimize the building efficiency, such as the choices of building materials, opening sizes, and glazing types. | The integration (Revit), VPL (Dynamo), and MOO (NSGA-II in MATLAB) facilitates the possibility of automating and speeding up the process of green retrofitting performance. |

**Table 4.** *Cont.*

| # | Author/Title | Objective | Methodology | Gap(s) | Contribution |
|---|---|---|---|---|---|
| 8 | Pruvost et al. [6]. Multimodel-based exploration of the building design space and its uncertainty | To support analysis of uncertainty by presenting an innovative modeling approach that collaborates building design workflow. | Integrating several disciplines to share information from different data models and formats to eventually be used as input in building energy analysis (BEA) including geometry, energy infrastructure, weather, building usage. | Computational methods lack rapid and more mature methods for designing options to find the best alternative. | The multimodal method is extended for a broad exploration of building design options and their inherent uncertainty. |
| 9 | Oti et al. [123]. Structural sustainability appraisal in BIM | To utilize API in BIM extension and demonstrates its application to embed sustainability issues in the structural conceptual design options in BIM. | The approach was achieved by mapping API for structural sustainability appraisal followed by developing assessment model and integrating this model using conceptual building design iterations. | APIs are not yet fully exploited in expanding the BIM scope. | The utilization process has expanded the BIM scope by modelling and creating algorithms applicable to enhance nD building performance. |
| 10 | Fuchs & Scherer [7]. Multimodels—Instant nD-modeling using original data | To introduce multimodels approach to offer wider value of information in terms of quality and time. | The loose cross-model coupling of data elements is neutrally stored in external ID-based link models. | The current interoperability lacks generality and satisfaction. | Offering multimodel approach to the single self-contained information space. |
| 11 | Cheng & Cheng [124]. Enhancing Multi-model Inference with Natural Selection | To employ the genetic algorithm (GA) to inspire the process of natural selection using crossover and mutation iteratively to update a collection of potential solutions (models) until convergence. | The use of the Markov chain theory to design an adaptive termination criterion that vastly reduces the computational cost. | The studies on the availability of candidate qualified models are very rare in literature. | It developed a new schema theory that characterizes the current model to improve the evolutionary process by demonstrating the GA empirical power based on two real data examples. |
| 12 | Deng et al. [81]. Research on safety management application of dangerous sources in engineering construction based on BIM technology | To create a construction hazard source safety management module through secondary development of the Revit platform. | Using the simulator Navisworks software to rescue the emergency of construction safety accidents by formulating corresponding emergency management plan | Existing hidden accidental dangers in construction without proper solution. | Introducing the security management module to guide developers to avoid accidents. |
| 13 | Li et al. [125]. Integration of Building Information Modeling and Web Service Application Programming Interface for assessing building surroundings in early design stages | To integrate Dynamo BIM and Amap web service APIs for the evaluations of diverse uses of transportations. | Results from the integrated tool are analyzed and validated with survey results | Developing service tools relates BIM and location to facilitate the process of estimating energy consumption. | The integration of Dynamo BIM and web service APIs is helpful for site assessments in the early design stage or even earlier. |
| 14 | Siegele et al. [127]. A new MATLAB Simulink Toolbox for Dynamic Building Simulation with BIM and Hardware in the Loop compatibility | To develop carnotUIBK for dynamic building simulations using MATLAB technique. | The development of this tool is carried out using programming the object-oriented language of MATLAB. | Research work using simulator MATLAB is very rare in BIM dynamic and loop compatibility. | The new model is tailored for hardware in the loop applications development and indoor air quality simulations in terms of multi-zone modelling. |
| 15 | Amiri et al. [95]. BIM-based applications of metaheuristic algorithms to support the decision-making process: Uses in the planning of construction site layout | To introduce BIM-based applications of metaheuristic optimization algorithms to support the decision-making process. | Metaheuristic optimization algorithms was employed by hybridizing several algorithms such as k-means approach. | Optimization has been treated inefficiently in most off the previous work especially in site layout planning. | BIM has been equipped with the optimized decision-making process by aggregating the necessary information at the right time. |

## 9. Contribution

Adopting algorithms in interface and IM in BIM multi-model applications for optimization has been considered the most significant contributor to achieving high performance in construction. This novel contribution was based on introducing suitable methodologies to accomplish new or upcoming research tasks. The process cannot be accomplished without specifying API in various software components that may interact, such as accessing the database, hard drive, disc drive, video card, etc. The interface is established to create programming codes equipped with programming language routines, data structures, and classes and variables. This review explains the integration of IBS and MBS in BIM and interfaces management. The other contribution of this study is to develop an API to enable BIM viewers to simplify BIM-based interface management.

One of the significant advanced steps to extend BIM use was developing a BIM-IoT system that allows one to directly use the BIM models for building context and 3D views. Alternatively, considering the BIM Model management in a native environment, VPL scripts have been developed to support the integration of BIM and IoT, which requires employing sensors.

VPL can also help introduce an accessible mode that enables BIM/IoT interfacing, in which BIM models could be transferred from static to dynamic, with the ability to self-update essential information. BIM/IoT opens a new trend of communication called machine-to-machine (M2M) communication, which extends to using external databases between the real and virtual worlds.

The review also presented an approach for a workflow engine to be integrated into the BIM multi-model collaboration platform. This type of implementation is considered the most critical step towards increasing the degree of automation. Another field, in addition to automation, was to create a multi-model BIM collaboration platform based on ontologies and BCF. The review has shown that the workflow of specific data can be obtained from the multi-model IDM/MVD methodology to achieve process and user model and their linkage information.

The review has introduced factors needed to implement better BIM, such as education, training, and studying BIM technology. In addition, BIM has expanded in scope as the algorithms were introduced to enhance nD building performance. Utilizing algorithms in BIM processes improved the process by demonstrating the GA empirical power based on two real data examples. Furthermore, integrating Dynamo BIM and web service APIs could improve site assessments in the early design stage, or even earlier.

## 10. Conclusions and Future Work

The review has shown clearly that there is still a long way to reach the objective of high performance of BIM. The interoperability issues were identified and reviewed; however, they need to be addressed in future versions of BIM tools schemas. The fundamental problem of these issues is slowing down the BIM automation and in areas that still employ manual interaction. The currently available tools likely can read, with minimum error, the BIM files. Most issues occur either when the BIM files are generated by CAD tools such as Revit or, more importantly, during reading the BIM files by energy simulation tools—a field that needs more focus and understanding in future research areas. The homogeneous data format can be easily used in the multi-model strategies instead of the generic data access to the elementary models, which requires creating a virtual structure throughout generalizing and idealizing the representation of popular data format concepts. Under this generalization, BIM applications can be extended without being conceptually changed, or maintaining a super data model.

The review has identified the possibility of utilizing the BACnet protocol for BAC information exchange that considers the IFC data model. Hence, the new technique can unlock the potential future of smart buildings using BIM and a BIM multi-model approach, which can be achieved by integrating tools and with BIM through IoT.

Despite the advancement of BIM in construction, the need for optimization BIM processes still represents the core of current and future developments. Recently, researchers have been working on adopting metaheuristic algorithms to support making decisions in construction by considering BIM-based applications. These applications include site layout to identify the size, energy consumption, and possible constraints concerning optimum cost outcomes. The algorithm's involvement has successfully enhanced productivity and safety in the construction process, saving cost and time by creating an intelligent system to control moving labor and materials. The optimization has shown clear evidence of the effectiveness of aggregating the necessary information at the right time.

The review presents limitations, especially in considering the potential link of BIM and Building Management System (BMS), and the level of influencing the BIM-IoT prototype. It was also revealed that the solution proposed by BIM skills to solve FM management in the dynamic model is still an unusual scenario because of the limited contribution of the BIM model in associating BMS environment.

The applications of interface management have been reviewed academically and practically in the literature, suggesting numerous BIM-based system developments in which CMI work is based on the filing system. This filing system has shortcomings which, alternatively, are replaced by proposing a DBCMI system, due to its capability to overcome the limitations in the other filing system. Employing a DBCMI system relies on establishing more effective visualization and sharing for BIM-based interface management during the project construction phase. It effectively helps integrate discourse in the BIM model and to improve the communication of information. Furthermore, designing API modules to be used in the DBCMI system simplifies using interface and operations that increase the willingness of participants to use the system. Despite the advancement of BIM in construction, the need to optimize BIM processes still represents the core of current and future developments.

The review presented the need for optimization by extending the contribution of the various algorithms in this process. Recently, researchers have been working on adopting metaheuristic algorithms to support deciding on construction by considering BIM-based applications. These applications include site layout to identify the size, energy consumption, and possible constraints concerning optimum cost outcomes. The algorithm's involvement has successfully shown an enhancement in productivity and safety in the construction process, which saves cost, time, and the framework of moving labor and materials. The optimization has shown clear evidence of the effectiveness of aggregating the necessary information at the right time. It was also revealed that the solution proposed by BIM skills to solve FM management in the dynamic model is still an unusual scenario because of the limitation of the BIM model contribution that is associated with the BMS environment.

**Author Contributions:** Conceptualization, N.A.H. (Nawal Abdunasseer Hmidah) and A.B.A.A.; methodology, N.A.H. (Nuzul Azam Haron); writing—original draft preparation, N.A.H. (Nawal Abdunasseer Hmidah), writing—review and editing, N.A.H. (Nuzul Azam Haron) supervision and reviewed the article, N.A.H. (Nuzul Azam Haron) advice and support, A.H.A. and T.H.L. and R.A.A.R.A.E. All authors have read and agreed to the published version of the manuscript.

**Funding:** This research received no external funding. This work was supported and partial funding by research management center, University Putra Malaysia.

**Institutional Review Board Statement:** Not applicable.

**Informed Consent Statement:** Not applicable.

**Data Availability Statement:** Not applicable.

**Acknowledgments:** The authors would like to thank the Department of Civil Engineering, Faculty of Engineering, University Putra Malaysia, Serdang, Malaysia, for supporting This review paper.

**Conflicts of Interest:** The authors declare no conflict of interest.

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
