# Peer review of "The Role of the Interface and Interface Management in the Optimization of BIM Multi-Model Applications: A Review"

_sustainability, doi:10.3390/su14031869_

Round 1

Reviewer 1 Report

The paper is framed as Systematic Review regarding the BIM interface, the BIM multi-model and the role of employing algorithms in BIM to achieve optimization.

Several novelties and contributions are pointed out but few are clearly a novelty.

General considerations:

The title is too long and should be shorten.

Sentences are often too long and get fuzzy. Confusing phrases mislead the readers understanding. This results in a poorly friendly reading paper. Also, significant English improvement is suggested. Proof-reading is encouraged.

At the end of the paper there are several issues missing/need to be edited. We highlight the identification of the author contributions.

Going in detail to more technical issues:

  • In the abstract it seems there is some confusion between formats and trademarks, as CAD is a document format and Revit we will assume that it is a trademark, meaning Autodesk REVIT software.
  • In the Introduction section the 1st sentence deserves some reflection namely when BIM is addressed as a tool. From the reading of the research authors assume BIM as multi-platform and multi-tool, where the introduction of algorithms is key for automation and higher ambition when using BIM. In this sense, BIM must be defined as a methodology gathering several processes and tools to improve projects and construction overall outcomes.
  • Regarding methodology:
    • As the PRISMA Statement emphasis, "A systematic review is a review of a clearly formulated question that uses systematic and explicit methods to identify, select, and critically appraise relevant research, and to collect and analyse data from the studies that are included in the review.".
    • The paper presents a lack of method. As no rigour methodology is presented, it is impossible to validate the “pseudo” review made. Based on that, it is not possible to validate the manuscript content, and the value of the work is unsupported.

In terms of development, given the scope it was expected to see tools that often support BIM on interfaces as EDMS - Electronic Document Management System or ALIM - Asset Lifecycle Information Management. Yet, no references are made to it.

The paper presents 13 Figures, where 11 are "inspired/copied" from other authors. Also, it seems that (based on the citation over the title) the 4 Tables presented 3 are from other authors. Although the respective authors are cited, the contribution of the work becomes compromised. Furthermore, since there is no methodology to support the choice of cited authors, the paper ends up being a collection of subjects selected by the authors.

In this respect, the energy issues appears without a solid justification.

Despite the high number of references, some relevant references, as example related to BIM Adoption are missing, as well as the group of ISO standards, ISO 19650 that clearly align the requirements, key elements and interfaces within BIM.

In resume, this paper needs significant improvement in terms of it structure, contents (methodology at the forefront) and proof-reading.

Author Response

REVIEWER #1

Comments and Suggestions for Authors

The paper is framed as Systematic Review regarding the BIM interface, the BIM multi-model and the role of employing algorithms in BIM to achieve optimization.

Several novelties and contributions are pointed out but few are clearly a novelty.

General considerations:

  1. The title is too long and should be shorten.

The current title: A Systematic Review for the Role of the Interface and Interface Management in Various BIM Multi-model Applications Through Algorithm Optimization (14 Words)

Modified Title: The Role of the Interface and Interface Management to Optimize the BIM Multi-Model Applications: A Review (10 Words)

  1. Sentences are often too long and get fuzzy. Confusing phrases mislead the readers understanding. This results in a poorly friendly reading paper. Also, significant English improvement is suggested. Proof-reading is encouraged.

The long sentences are shortened. English is improved.

  1. At the end of the paper there are several issues missing/need to be edited. We highlight the identification of the author contributions.

Going in detail to more technical issues:

  1. In the abstract it seems there is some confusion between formats and trademarks, as CAD is a document format and Revit, we will assume that it is a trademark, meaning Autodesk REVIT software.

Original: The challenge that faces adopting BIM lies in the limiting ability of the computer-aided design (CAD) and Revit to generate a simple procedure to be read by BIM requiring homogeneous data format to generalize better and maintain a super data model.

The challenge that faces adopting BIM lies in the limiting ability of the computer-aided design (CAD) to generate a readable and straightforward Revit by BIM requiring homogeneous data format to be generalized better and maintain a super data model.

  1. In the Introduction section the 1st sentence deserves some reflection namely when BIM is addressed as a tool. From the reading of the research authors assume BIM as multi-platform and multi-tool, where the introduction of algorithms is key for automation and higher ambition when using BIM. In this sense, BIM must be defined as a methodology gathering several processes and tools to improve projects and construction overall outcomes.

Original: BIM is a project-improving tool that globally provides a revolutionary platform for designing, construction, maintaining, operating, and improving various fields of rehabilitation, retrofit, and redevelopment to existing assets in the built environment.

BIM is a project-improving tool that globally provides a revolutionary platform for designing, construction, maintaining, operating, and improving various fields of rehabilitation, retrofit, and redevelopment to existing assets in the built environment. Another helpful definition considers BIM as a methodology gathering several processes and tools to improve projects and construction overall outcomes.

  1. Regarding methodology: As the PRISMA Statement emphasis, "A systematic review is a review of a clearly formulated question that uses systematic and explicit methods to identify, select, and critically appraise relevant research, and to collect and analyze data from the studies that are included in the review." The paper presents a lack of method. As no rigor methodology is presented, it is impossible to validate the “pseudo” review made. Based on that, it is not possible to validate the manuscript content, and the value of the work is unsupported.

Response:  I agree that PRISMA statement is correct.  This review has devoted a complete section to discuss previous methodologies (Section 6) which includes integrating CMI (6.1) and integrating IM (6.2).  These two methodological elements were considered as the fundamental of the integrating multi-model BIM which is the core of the paper. 

  1. In terms of development, given the scope it was expected to see tools that often support BIM on interfaces as EDMS - Electronic Document Management System or ALIM - Asset Lifecycle Information Management. Yet, no references are made to it.

Response: I agree with the reviewer that including EDMS and ALIM will enhance the methodology contents.  For this reason, Section 6.3 (Document Management System) has been added as follows:

6.3 Document Management System

Reference #3

The complete knowledge about the building material and corresponding repair and condition information are provided in the ‘SharePoint’ electronic document management system (EDMS).  The most crucial benefit of EDMS is that storing information can be done without a formal structure. The information becomes available and can be obtained at any time to be used as needed.  Recently, the EDMS information has been stored with some structure to the documents to transfer information easily.  The requirement of the information could become a computable BIM data parameter and enable reviewing the existing template documents (Hull, J., & Ewart, I. J. (2020). Conservation data parameters for BIM-enabled heritage asset management. Automation in Construction, 119, 103333).

Reference #4

Regarding the Asset Lifecycle Information Management (ALIM), BIM could benefit from the open standards that enable future builders and developers to record information on the entire lifecycle to be managed.  The importance of ALIM arises because of the availability of information as an essential part of structural digital twins.  This type of system provides detailed insights into performance and processes through simulation. This information can be combined and reused quickly if specified, designed, realized, managed, maintained, and dismantled according to open standards.  Linking and connecting data are innovative concepts since ALIM requires bringing together diverse sources from different domains with their standards; knowledge of various standards is essential. Therefore, ALIM provides extensive expertise on both data integration and data modelling based on linked data and (inter)national open standards (Wang, W., Hu, H., Zhang, J., & Hu, Z. (2020, December). Digital Twin-based Framework for Green Building Maintenance System. In 2020 IEEE International Conference on Industrial Engineering and Engineering Management (IEEM) (pp. 1301-1305). IEEE).

  1. The paper presents 13 Figures, where 11 are "inspired/copied" from other authors. Also, it seems that (based on the citation over the title) the 4 Tables presented 3 are from other authors. Although the respective authors are cited, the contribution of the work becomes compromised. Furthermore, since there is no methodology to support the choice of cited authors, the paper ends up being a collection of subjects selected by the authors.

  1. In this respect, the energy issues appear without a solid justification.

Response: It is true since the review papers are normally “collecting data” from various sources.  The paper has devoted Section 5.2 (Framework for Energy Management) to justify the presence of the techniques depicted in Figure 7 (The mechanism of interface management system [32]). 

  1. Despite the high number of references, some relevant references, as example related to BIM Adoption are missing, as well as the group of ISO standards, ISO 19650 that clearly align the requirements, key elements and interfaces within BIM.

Response:  Some references have been added to cover these areas.

  1. In resume, this paper needs significant improvement in terms of its structure, contents (methodology at the forefront), and proofreading.

Response:  Thanks. Absolutely, the paper is enhanced.

Reviewer 2 Report

General comments:

The subject of the proposed manuscript is of great interest, analyzing several of the most relevant topics of the BIM methodology in the AEC sector. However, the reviewer considers that the manuscript lacks in the form and structure of the document. The sensation that the reader has is the absence of a clear common thread that allows to connect and integrate the different topics analyzed (Interface, IFC, Frameworks,…). This aspect may be determined because the authors have not clearly defined the objective pursued by the research. What is the Research Question? Nor are the existing challenges in the matter and that should be addressed in a clear way.

Furthermore, considering that the proposed manuscript is a “Systematic Review (SR)”, the reviewer understands that the manuscript must reflect the methodology carried out for the literature review used and this aspect has not been developed. The tools used from databases of scientific literature are not presented, nor are the sequence used for filtering and selection criteria of articles considered or other fundamental analyzes in SR-type articles. I recommend that authors consider the Sustainability Journal guidelines set out in the Instructions for Authors section (https://www.mdpi.com/journal/sustainability/instructions), specifically the “Types of Publications” section where the guide lines are detailed of PRISMA for SR type documents. I also recommend, by way of example, to consider other studies of the SR type to know the typical structures of this type of manuscripts (eg https://doi.org/10.1016/j.autcon.2021.103642, https://doi.org/ 10.3390 / buildings11080336, https: //doi.org/10.1016/j.autcon.2021.103760 ...)

The reviewer recommends including in the introduction section some content that allows the reader to encompass the framework within the general theme to which the research belongs (Digitization of the construction sector, new methodologies for the management of construction projects, ...) . The authors start directly with the BIM topic and adding a paragraph of the general framework would facilitate the creation of a common thread.

On the other hand, the abstract, like the introduction section, lacks a couple of lines that encompass the framework where this research is located. He begins by detailing the specific topics covered in the article without offering an initial overview. The reviewer proposes that the authors follow the Journal's recommendations described on the website.

Finally, I recommend that authors pay attention to the originality when writing manuscripts. The use of certain phrases of other authors is understandable if they are cited and referenced accordingly (especially in the introductory sections). However, several sections have been detected with the same written content  as other available documents of other authors and that have not been cited, and the feeling it gives the reader is that they are original paragraphs of the authors of this manuscript but that in reality they are not.

Minor comments

- Line 142, MI correct to IM

- Line 153-155 “… BIM parametric 3D computer-aided design (known as AEC) refers to parametric 3D CAD techniques and processes. … ”AEC refers to Architecture, Engineering and Construction not computer-aided design (CAD).

- Insufficient image quality of Figure 3 and Figure 8. Many parts are blurred.

- Line 220 and 521, "Ips" correct for “IPs”

- Line 385. “… integrating BEMS in BMI by creating a…” Correct BMI by BIM

- Line 499. Title "Figure 10. Investigation of CMI and BMI model [34]." Correct BMI by BIM

- Line 809: "Recently6y"

- Line 837-857 repeats the same content as described in lines 802-822

- I propose that the author consider the following investigations related to the Section 5.2. Framework for Energy Management:

o Related to Energy Management issues and automated 3D modeling (https://doi.org/10.3390/buildings11090380)

o Related to BIM and IoT integration (https://doi.org/10.1016/j.autcon.2019.01.020;  https: //doi.org/10.1016/j.autcon.2018.07.022,

Author Response

REVIEWER #2

Comments and Suggestions for Authors

General comments:

  1. The subject of the proposed manuscript is of great interest, analyzing several of the most relevant topics of the BIM methodology in the AEC sector. However, the reviewer considers that the manuscript lacks in the form and structure of the document. The sensation that the reader has is the absence of a clear common thread that allows to connect and integrate the different topics analyzed (Interface, IFC, Frameworks,….). This aspect may be determined because the authors have not clearly defined the objective pursued by the research. What is the Research Question? Nor are the existing challenges in the matter and that should be addressed in a clear way.

Response: Thanks.  I would like to refer the respected reviewer to the last sentence of the Introduction: “linking existing libraries where huge information about the thermal conductivity properties is available.  The life cycle assessment of a building can be estimated better by integrating CAD and BIM. This link provides information about optimizing the building envelope or sizing the HVAC system [21].”  The objective of the paper has been clearly identified. 

The challenges to achieve such an objective is mentioned “Interface Management (IM) is another concern of interface developed to address the challenges of managing complex capital projects to face the rising complexity due to globalization and the geographical distribution of various cultures [24]” and “potential challenges posed by the visual programming languages (VPLs)”. 

  1. Furthermore, considering that the proposed manuscript is a “Systematic Review (SR)”, the reviewer understands that the manuscript must reflect the methodology carried out for the literature review used and this aspect has not been developed. The tools used from databases of scientific literature are not presented, nor are the sequence used for filtering and selection criteria of articles considered or other fundamental analyzes in SR-type articles. I recommend that authors consider the Sustainability Journal guidelines set out in the Instructions for Authors section (https://www.mdpi.com/journal/sustainability/instructions), specifically the “Types of Publications” section where the guidelines are detailed of PRISMA for SR type documents. I also recommend, by way of example, to consider other studies of the SR type to know the typical structures of this type of manuscripts (eg

https://doi.org/10.1016/j.autcon.2021.103642,

https://doi.org/ 10.3390 / buildings11080336,

https: //doi.org/10.1016/j.autcon.2021.103760 ...)

Response:  I agree with the reviewer that the style of the systematic review has not been made.  Accordingly, the word “systematic” has been removed from the title.

  1. The reviewer recommends including in the introduction section some content that allows the reader to encompass the framework within the general theme to which the research belongs (Digitization of the construction sector, new methodologies for the management of construction projects, ...) . The authors start directly with the BIM topic and adding a paragraph of the general framework would facilitate the creation of a common thread.

Response: The flowchart of how the paper is constructed was presented in Figure 1.  The section follows the chart presents the fundamental information that is needed to explain the following sections.  It is true that the framework presented were taken from related sources; however, optimization was discussed later in Section 5. 

  1. On the other hand, the abstract, like the introduction section, lacks a couple of lines that encompass the framework where this research is located. He begins by detailing the specific topics covered in the article without offering an initial overview. The reviewer proposes that the authors follow the Journal's recommendations described on the website.

[Note-01]: Response: This part is to be added to the Abstract: The enhancement of BIM could be performed by involving algorithms to achieve better results in productivity, safety, cost, time, and constructing framework. 

  1. Finally, I recommend that authors pay attention to the originality when writing manuscripts. The use of certain phrases of other authors is understandable if they are cited and referenced accordingly (especially in the introductory sections). However, several sections have been detected with the same written content as other available documents of other authors and that have not been cited, and the feeling it gives the reader is that they are original paragraphs of the authors of this manuscript but that in reality they are not.

Response: Care is taken when the citation is done. 

  1. Minor comments

- Line 142, MI correct to IM

Response: Corrected

- Line 153-155 “… BIM parametric 3D computer-aided design (known as AEC) refers to parametric 3D CAD techniques and processes. … ”AEC refers to Architecture, Engineering and Construction not computer-aided design (CAD).

Response: The words “(known as AEC)” were removed.

- Insufficient image quality of Figure 3 and Figure 8. Many parts are blurred.

Response: Both Figures were regenerated.

- Line 220 and 525, "Ips" correct for “IPs”

Response; Both corrected

- Line 385. “… integrating BEMS in BMI by creating a…” Correct BMI by BIM

Response: Corrected

- Line 499. Title "Figure 10. Investigation of CMI and BMI model [34]." Correct BMI by BIM

Response: Corrected

- Line 809: "Recently6y"

Response: Corrected

- Line 837-857 repeats the same content as described in lines 802-822

Response: All removed

  1. I propose that the author consider the following investigations related to the Section 5.2. Framework for Energy Management:

o Related to Energy Management issues and automated 3D modeling (https://doi.org/10.3390/buildings11090380)

Reference #1

There is another approach to generate a dynamic energy simulation model for a single existing building whose aim is to collecting existing data to prepare energy retrofits at the lowest cost possible.  The proposal includes establishing a polygon model by employing photogrammetrically generated point clouds with the Tool for Energy Analysis and Simulation for Efficient Retrofit (TEASER) and AixLib. A single-family house was taken as a case study to achieve the purpose.  The model reproduces the internal air temperatures during synthetical heating up and cooling down with building heat transfer coefficients (HTC) agree within a 12% range. The model requires accurate window characterisations and justifies the use of a very simplified interior geometry. However, uncertainties arose regarding comparing different typologies showing differences in pre-retrofit heat demand of about ±20% to the average.

[Gorzalka, P., Estevam Schmiedt, J., Schorn, C., & Hoffschmidt, B. (2021). Automated Generation of an Energy Simulation Model for an Existing Building from UAV Imagery. Buildings, 11(9), 380.]

o Related to BIM and IoT integration (https://doi.org/10.1016/j.autcon.2019.01.020;  https: //doi.org/10.1016/j.autcon.2018.07.022,

Reference 2

The other application is about the modern high-rise buildings, which requires a diverse building environment with multiple variables that make fire hazards challenging to predict and monitor accurately. To avoid any possible tragedy, creating, developing, and implementing an integrated fire disaster prevention system has become necessary to prevent fire disasters and adequately protect life and property effectively. The disaster response system includes a complete plan of evacuation planning and rescue guidance.  The drawback of the disaster system is that it is prone to error due to the inaccuracy, incompleteness, and poor communication of this intelligence. However, the role of BIM is to minimize possible human errors.  The system could be better performed in case of providing 3D visualization to support the assessment, planning, and detection of fire safety.

[Cheng, M. Y., Chiu, K. C., Hsieh, Y. M., Yang, I. T., Chou, J. S., & Wu, Y. W. (2017). BIM integrated smart monitoring technique for building fire prevention and disaster relief. Automation in Construction, 84, 14-30.}

Reviewer 3 Report

The article performs a systematic review of the role of the interface and its management in various BIM multi-model applications using algorithms. In general, the article addresses a very current topic, since it analyzes the methodologies used to optimize the automation of processes to avoid errors or information loss in the information exchange between different agents. However, the article does not seem to have an adequate balance when presenting concepts: the part where concepts are introduced and defined ( first 8 points) is too extensive. It could be compressed and / or reduced. There are concepts or ideas that are repeated in several sections. On the other hand, the contribution section is very brief.

Besides, there are some acronyms that are discussed before defining them. For example, IMS is commented on line 200 and it is defined later on line 217. Figure 2 is commented on lines 196-202,. This figure shows some items (IAI) that are not defined until table 2 (line 229) and they are not even commented in the figure description.

Figures 2 and 5 are very similar. Furthermore, Figure 5 provides more information than Figure 2., so it could be used both to describe the IM components and the IME hierarchy without adding Figure 2.

Author Response

REVIEWER #3

Comments and Suggestions for Authors

  1. The article performs a systematic review of the role of the interface and its management in various BIM multi-model applications using algorithms. In general, the article addresses a very current topic, since it analyzes the methodologies used to optimize the automation of processes to avoid errors or information loss in the information exchange between different agents. However, the article does not seem to have an adequate balance when presenting concepts: the part where concepts are introduced and defined (first 8 points) is too extensive. It could be compressed and / or reduced. There are concepts or ideas that are repeated in several sections. On the other hand, the contribution section is very brief.

Response: There are two points here.  Reducing the concept section and stretching the contribution section.  At least one of the other reviewers asked to enlarge the concept section, so, I prefer to keep it as is.  Regarding the contribution section, I will improve it. 

[ADD]

The review has introduced factors needed to implement better BIM, such as education, training, and studying BIM technology.  In addition, BIM has expanded in scope as the algorithms were introduced to enhance nD building performance.  Utilizing algorithms in BIM processes improved the process by demonstrating the GA empirical power based on two real data examples.  Further, integrating Dynamo BIM and web service APIs could improve site assessments in the early design stage or even earlier.

  1. Besides, there are some acronyms that are discussed before defining them. For example, IMS is commented on line 200 and it is defined later on line 217. Figure 2 is commented on lines 196-202,. This figure shows some items (IAI) that are not defined until table 2 (line 229) and they are not even commented in the figure description.

Response:

IMS on line 200 was moved to line 219.

IAI is explained.

  1. Figures 2 and 5 are very similar. Furthermore, Figure 5 provides more information than Figure 2., so it could be used both to describe the IM components and the IME hierarchy without adding Figure 2.

Response: Figure 2 is devoted to mention the components.  Figure 5 is devoted to show the hierarchy of these components.

Reviewer 4 Report

e paper reviews the role of interface and interface management in various BIM multimodel applications. There are some main issues that should be addressed before being suitable for publishing the paper.

Authors should review the other papers that have worked on the building information modelling and its application to highlight existing review papers in the body of the literature and then highlight the importance of their work in comparison to these existing review papers. However, authors have only talked about the BIM, the importance and the application of the BIM without mentioning and reviewing these literature papers that have already covered the concept of building information modelling. They should clearly highlight what is the difference between their work to these papers and what will be the contribution of this paper.

This paper requires a methodology section that shows how the literature review is organized and conducted, how the papers are selected, how many papers are they have reviewed, what was the specific methodology to read the papers, categorize them, extract information and create the concepts that are presented in Figure 1.

The information provided in Section 8 is irrelevant to the structure of the paper. This section can be removed and the information provided in this section can be integrated with previous sections to show the examples of the concepts and algorithms that are introduced in previous sections or if authors prefer to keep this section, this section should highlight the gaps, issues or opportunities that are extracted from the empirical research. In this version, it only shows a summary of empirical research that can be integrated into previous sections.

In table fou,r it's not clear that the gaps are the gaps of that specific paper and authors by reviewing that paper has found this gap or this is the gap that the study is trying to solve in that specific paper, in this way the gaps should have been addressed in the next column or contribution. The methodology section in this table should be summarized and only shows the main approach and method that this paper has used.

Authors should review the paper in terms of the grammatical errors and minor issues for example look at column 5 row 1.

Author Response

REVIEWER #4

Comments and Suggestions for Authors

e paper reviews the role of interface and interface management in various BIM multimodel applications. There are some main issues that should be addressed before being suitable for publishing the paper.

  1. Authors should review the other papers that have worked on the building information modelling and its application to highlight existing review papers in the body of the literature and then highlight the importance of their work in comparison to these existing review papers. However, authors have only talked about the BIM, the importance and the application of the BIM without mentioning and reviewing these literature papers that have already covered the concept of building information modelling. They should clearly highlight what is the difference between their work to these papers and what will be the contribution of this paper.

Response: The organization of the paper is explained in Figure 1 as a reflection to the title of the paper.  The information gathered from various sources reflects the title and corresponding organization presented in Figure 1.  The topic is slightly huge owing to the nature of the review itself.  For this reason, it is very hard to combine all ideas at certain locations. However, some changes have been performed according to opinions of other reviewers.  Hope that is enough to answer this concern. 

  1. This paper requires a methodology section that shows how the literature review is organized and conducted, how the papers are selected, how many papers are they have reviewed, what was the specific methodology to read the papers, categorize them, extract information and create the concepts that are presented in Figure 1.

Response: Actually, Section 5 (Developing Framework) is meant to be the Methodology.  In this section, all details can be found that may address the concern here.

  1. The information provided in Section 8 is irrelevant to the structure of the paper. This section can be removed and the information provided in this section can be integrated with previous sections to show the examples of the concepts and algorithms that are introduced in previous sections or if authors prefer to keep this section, this section should highlight the gaps, issues or opportunities that are extracted from the empirical research. In this version, it only shows a summary of empirical research that can be integrated into previous sections.

Response:  By looking at the objective of each of the papers presented in Section 8, one can say:

  1. Mukkavaara: support automation within a BIM-based design process. [Related]
  2. Eray et al.: To develop a framework for integrating IM and BIM systems to create better coordination. [Related]
  3. Tang et al. : To link smart buildings to different building systems together with the Building Automation System (BAS). [Related]
  4. Lin et al. : To integrate BIM and web technology.[Related]
  5. Kang: To introduce Building Information Modeling (BIM)-based Human Machine Interface (HMI) framework for intuitive space-based energy management. [Related]
  6. Leśniak et al.: Implementing BIM technology in the construction process in Poland. [Related]
  7. Lim et al.: To automate the computational building information modelling (BIM) in decision-making for green retrofitting of the existing building. [Related]
  8. Pruvost et al.: presenting an innovative modeling approach that collaborates building design workflow. [Indirectly Related]
  9. Oti et al.: To utilize API in BIM extension and demonstrates its application to embed sustainability issues. [Related]
  10. Fuchs & Scherer: To introduce multimodels approach to offer wider value of information in terms of quality and time. [Related]
  11. Cheng & Cheng: To employ the genetic algorithm (GA). [Related]
  12. Deng et al.: To create a construction hazard source safety management module through secondary development of the Revit platform. [Related]
  13. Li et al.: To integrate Dynamo BIM and Amap web service APIs for the evaluations of diverse uses of transportations. [Related]
  14. Siegele et al.: To develop carnotUIBK for dynamic building simulations using MATLAB technique. [Related; Optimkization]
  15. Amiri et al.; To introduce BIM-based applications of metaheuristic optimization algorithms to support the decision-making process. [Related; Algorithm]

  1. In table four it's not clear that the gaps are the gaps of that specific paper and authors by reviewing that paper has found this gap or this is the gap that the study is trying to solve in that specific paper, in this way the gaps should have been addressed in the next column or contribution. The methodology section in this table should be summarized and only shows the main approach and method that this paper has used.

Response: The gap(s) presented in Table 4 reflects the gaps in the corresponding papers. 

Secondly, the Summary briefly describes the papers according to some criteria such as gap, objectives, and others.  Also, shows a discrimination amongst these papers. 

  1. Authors should review the paper in terms of the grammatical errors and minor issues for example look at column 5 row 1.

Response: Agree.

Round 2

Reviewer 1 Report

The paper is framed as Systematic Review. However, considering all the comments and revisions it results at the most on a Review. Therefore it is recommended to re-frame this paper.

General considerations:

The title was arranged and expresses better the aim.

Significant improvement was performed in English writing. However there are still some minor errors that must be amended.

One key aspect that was not corrected is at the end of the paper. There are still several issues missing/need to be edited. I continue to highlight the identification of the author contributions.

Figure 1 still mentions “Systematic Review”. This should be corrected.

Going in detail to more technical issues:

  • There is still some confusion on the abstract between formats and trademarks, as CAD or REVIT. The corrections made did not help on clearing the understanding.
  • Regarding methodology, the paper still presents a lack of method.

All figures were maintained, where 11 are from other authors. The still paper looks as a collection of subjects selected by the authors where few contributions result from their sequence of relevance for the aim of the research.

Despite the significant improvements there are still relevant aspects that must be improved.

Author Response

Reviewer 1

Comments and Suggestions for Authors

The paper is framed as a Systematic Review. However, considering all the comments and revisions it results at the most on a Review. Therefore, it is recommended to re-frame this paper.

Response: I agree with the reviewer that the paper must be turned into a “review” rather than a “systematic” review paper. However, from the beginning, there was a mistake. The paper was written as a “review” paper, and it was framed that way. Based on the requirements of the “review” paper, the authors would like to highlight the most important features of the “review” paper with the compliance of the paper: 

  • Review papers provide an overview of what is known about a particular topic. (Comply)
  • Review papers evaluate the material, rather than simply restating it, but the methods used to do this are not usually prespecified, and they are not described in detail in the review. (Comply)
  • Review papers might be comprehensive, but it does not aim to be exhaustive. Literature reviews are also referred to as narrative reviews. (Comply)
  • Review papers use a topical approach and often take the form of a discussion. Precision and replicability are not the focus; rather the author seeks to demonstrate their understanding and perhaps also present their work in the context of what has come before. (Comply)

All the requirements (features) of the “review” paper are there.

General considerations:

The title was arranged and expresses better the aim.

Response: Thanks

Significant improvement was performed in English writing. However, there are still some minor errors that must be amended.

Response: The whole paper has reviewed

One key aspect that was not corrected is at the end of the paper. There are still several issues missing/need to be edited. I continue to highlight the identification of the author contributions.

Figure 1 still mentions “Systematic Review”. This should be corrected.

Response: Corrected

Going in detail to more technical issues:

There is still some confusion on the abstract between formats and trademarks, as CAD or REVIT. The corrections made did not help on clearing the understanding.

Response:  A trademark is a word, name, symbol, or device, or combination of these elements, which identifies the goods or services and distinguishes those goods from the goods of another.  Examples are ADSK®, AutoCAD LT®, BIM 360®.  On the other side, formats support a wide range of industry standards and file formats.

Examples of formats are RVT, RFA, RTE, RFT. CAD formats: DGN, DWF, DWG, DXF, IFC, SAT, and SKP. Image formats: BMP, PNG, JPG, JPEG.

Honestly, I do not figure out how the “formats and trademarks” have become a matter of confusion. 

Regarding methodology, the paper still presents a lack of method.

Response: The purpose of the Review paper is NOT to present a methodology. Instead of searching for a few methodologies and aligning them based on the title (objective) of the paper is "the Role of the Interface and Interface Management to Optimize the BIM Multi-Model Applications: A Review." The paper's objective is to search for previous methodologies that propose a complete methodology or partial contribution. This paper aims to gather any possible and related information to establish a logical method.

Reviewing previous related methods is shown in Section 6. The sub-sections are 6.1 Integrating CMI, 6.2 Integrating MI, and 6.3 Document Management System. This part contributes to forming a BIM multi-model by integrating CMI and MI, as shown in Figure 10 and Figure 11. The last section (6.3) presents the documentary part of the integration towards the BIM multi-model.

The second part of the tile can be achieved by Section 7, which thoroughly proposes information about the BIM-base algorithm. 

The methodology can be seen throughout these two sections

All figures were maintained, where 11 are from other authors. The still paper looks as a collection of subjects selected by the authors where few contributions result from their sequence of relevance for the aim of the research.

Response: This is the nature of the Review Papers: Collection, Arranging, and Reflection.  The current paper is different from other papers in this direction.

Despite the significant improvements there are still relevant aspects that must be improved.

Reviewer 2 Report

As you can see, I used the Turnitin tool which allowed me to see a lot of content very similar to other research papers and articles throughout the manuscript. Although there is much content similar to other works throughout the document, I have highlighted some paragraphs marked by a green box and added the source as an annotation. On the other hand, the figures used are also very similar to others used in the articles of origin.

The general feeling that I have had when reading the article is lack of originality and that a lot of content is literally obtained from other articles.

Author Response

Reviewer 2

Comments and Suggestions for Authors

As you can see, I used the Turnitin tool which allowed me to see a lot of content very similar to other research papers and articles throughout the manuscript. Although there is much content similar to other works throughout the document, I have highlighted some paragraphs marked by a green box and added the source as an annotation. On the other hand, the figures used are also very similar to others used in the articles of origin.

The general feeling that I have had when reading the article is lack of originality and that a lot of content is literally obtained from other articles.

Response: There are indeed similar contents in both the text and Figures.  The Turnitin without involving references is about 19% similar.  Based on your highlight parts of similarity.  I went through these parts and rephrased them differently.  I hope the new version is satisfying you. 

Concerning the lack of originality, I humbly say that gathering such huge information from 128 references, arranging this information, and proposing a title is in itself “is an original act.”  Look at the contribution of the study and the informative quantity of knowledge that clearly shows the possibility of developing approaches to implement specific algorithms to optimize the activities of BIM in the industry. 

Reviewer 4 Report

It can be accepted in this verison

Author Response

thanks.